# Integration of mouse ovary morphogenesis with developmental dynamics of the oviduct, ovarian ligaments, and rete ovarii

Jennifer McKey[1]*, Dilara N Anbarci[1], Corey Bunce[1], Alejandra E Ontiveros[2], Richard R Behringer[2], Blanche Capel[1]*

[1]Department of Cell Biology, Duke University Medical Center, Durham, United States; [2]Department of Genetics, The University of Texas MD Anderson Cancer Center, Houston, United States

**Abstract** Morphogenetic events during the development of the fetal ovary are crucial to the establishment of female fertility. However, the effects of structural rearrangements of the ovary and surrounding reproductive tissues on ovary morphogenesis remain largely uncharacterized. Using tissue clearing and lightsheet microscopy, we found that ovary folding correlated with regionalization into cortex and medulla. Relocation of the oviduct to the ventral aspect of the ovary led to ovary encapsulation, and mutual attachment of the ovary and oviduct to the cranial suspensory ligament likely triggered ovary folding. During this process, the rete ovarii (RO) elaborated into a convoluted tubular structure extending from the ovary into the ovarian capsule. Using genetic mouse models in which the oviduct and RO are perturbed, we found the oviduct is required for ovary encapsulation. This study reveals novel relationships among the ovary and surrounding tissues and paves the way for functional investigation of the relationship between architecture and differentiation of the mammalian ovary.

*For correspondence:
mckey.jennifer@gmail.com (JM);
blanche.capel@duke.edu (BC)

**Competing interest:** The authors declare that no competing interests exist.

## Editor's evaluation

This work presents a very detailed study on the morphological processes governing mouse ovary morphogenesis from E14.5 to birth using recently developed methods. As a reference to ovary morphogenesis and the establishment of the female genital tract, the manuscript is of interest to all developmental biologists as it will serve as a reference to whole embryo morphogenesis, in particular vertebrate ovary morphogenetics processes.

## Introduction

During mouse fetal development, the gonad first arises on the ventromedial aspect of the mesonephros (*Brambell and Hill, 1927*; *Harikae et al., 2013*; *Karl and Capel, 1998*; *Windley and Wilhelm, 2015*). In XX embryos, the gonads develop into ovaries, where supporting cells become granulosa cells, interstitial cells become theca cells and other stromal cell types, and germ cells become oocytes (*Nef et al., 2019*; *Rastetter et al., 2014*; *Stévant et al., 2019*; *Wilhelm et al., 2013*). As these cells undergo differentiation between E11.5 and birth, they progressively assemble into individual ovarian follicles, characterized by the presence of one oocyte surrounded by two or three pre-granulosa cells (*Lei and Spradling, 2016*; *Lei and Spradling, 2013*; *Liu et al., 2010*; *Mork et al., 2012*; *Pepling and Lei, 2018*; *Pepling, 2012*; *Zheng et al., 2014*).The advent of single-cell transcriptomics has provided

**eLife digest** In humans and other mammals, the female reproductive organs, or ovaries, develop early in life, while the young are still in their mother's womb. Ovaries contain several different compartments, including the ovarian follicles. These are small groups of cells that produce reproductive hormones, and each follicle also has the potential to produce one egg for fertilisation. The ovaries are further surrounded by different tissues that develop alongside them. These include the oviducts, which carry fertilised eggs from the ovaries into the womb, and ligaments, which anchor the ovaries to the wall of the body cavity.

During the development of ovaries, ovarian follicles are sorted into two distinct groups. The first, called medullary follicles, are lost before puberty. The second group, or cortical follicles, remain in a state of 'suspended animation' until puberty. After that, they act as a 'reserve' of eggs for the rest of the reproductive lifespan. Once each cortical follicle has produced an egg, it is not replenished. This means that proper follicle sorting is crucial for establishing female fertility, and therefore the ability to conceive. The mechanisms behind follicle sorting, however, are still poorly understood.

McKey et al. set out to determine how the ovary's structure changed during its development. In the experiments, high-resolution microscopy techniques were used to reconstruct ovaries of mice in 3D across different stages of development.

This revealed that the ends of each ovary started folding towards each other just before birth, and that the folding also happened at the same time as follicle sorting. Simultaneous changes in the shape and orientation of the ligaments suggested that these tissues might direct the folding, for example by pushing or pulling on the rest of the ovary.

These results suggest that the changes in ovary structure in early life are critically linked to the establishment of the ovary's egg reserves. McKey et al. hope that this study will pave the way to a better understanding of infertility and, ultimately, better treatments.

a wealth of information on the cellular profiles and gene expression trajectories within the developing ovary, presenting an unprecedented view of the molecular and cellular pathways at play during mammalian ovary differentiation and follicle formation (*Li et al., 2017*; *Liu et al., 2010*; *Mayère et al., 2022*; *Niu and Spradling, 2020*; *Stévant et al., 2019*). However, transcriptomics alone do not address the spatial and structural components that underlie ovary regionalization and patterning. In fact, while granulosa cells and oocytes are locally assembling into follicles, the ovary as a whole undergoes dramatic changes in shape and cell composition, including the integration of extrinsic components such as lymphatics, mesenchymal theca cells, and innervation (*Liu et al., 2015*; *McKey et al., 2019*; *Svingen et al., 2012*). The events that occur during ovary morphogenesis are poorly defined and lack context, owing in part to the scarcity of molecular and imaging tools to study the ovary in its native position. Similarly, tissues closely associated with the ovary, such as the Müllerian duct (MD), ovarian ligaments, and the rete ovarii (RO), that also undergo dramatic transitions during development *Adham et al., 2002*; *Ford et al., 2021*; *Lee et al., 2011*; *Wenzel and Odend'hal, 1985* have been studied individually, but not in the context of ovary morphogenesis. Consequently, their impact on ovary morphogenesis has been overlooked. With recent advances in tissue clearing and 3D imaging technology, it is now possible to visualize structural changes at the level of the whole gonad or even the whole embryo (*Bunce et al., 2021*; *Fiorentino et al., 2021*; *Soygur et al., 2021a*; *Soygur and Laird, 2021b*). We recently developed a tissue-clearing method that combines CUBIC and iDISCO (*McKey et al., 2020*; *Renier et al., 2014*; *Susaki et al., 2015*), and optimized 3D imaging methods using lightsheet microscopy to visualize the fetal and perinatal ovary in situ, in the context of its surrounding tissues (*McKey et al., 2019*).

Using our 3D-imaging methods and high-resolution confocal microscopy, we provide a detailed and contextual description of mouse ovary morphogenesis from E14.5 to birth. We define three discrete hallmarks of ovary morphogenesis: (1) ovary folding, (2) specification of medullary and cortical compartments, and (3) ovary encapsulation. We integrated this morphometric approach with a detailed study of three tissues closely associated with the ovary: the ovarian ligaments, the MD and the RO, and found compelling correlations between the developmental dynamics of these tissues and ovary morphogenesis. Finally, we found that different combinations of *Pax2* and *Pax8* deletion

alleles perturb the oviduct and/or regions of the RO differentially, and lead to disruptions in ovary morphogenesis.

## Results

### 3D models of developing ovaries provide a new perspective on ovary morphology

The gonad first arises on the coelomic surface of the mesonephros. Early gonads are generally visualized and studied in complex with the mesonephros, which hosts the developing reproductive ducts. The mesonephric duct, which gives rise to the Wolffian duct (WD), is present from the onset of gonadogenesis (*Brambell, 1927*; *Kobayashi and Behringer, 2003*; *Zhao and Yao, 2019*). However, the MD develops later by invagination of the mesonephric epithelium between E11.5 and E13.5 in the mouse (*Huang et al., 2014*; *Mullen and Behringer, 2014*; *Orvis and Behringer, 2007*; *Roly et al., 2018*). While both WD and MD initially develop in all embryos, only one is maintained depending on the sex of the embryo. In female embryos, the WD fully regresses by E15.5 (*Zhao et al., 2021*), while

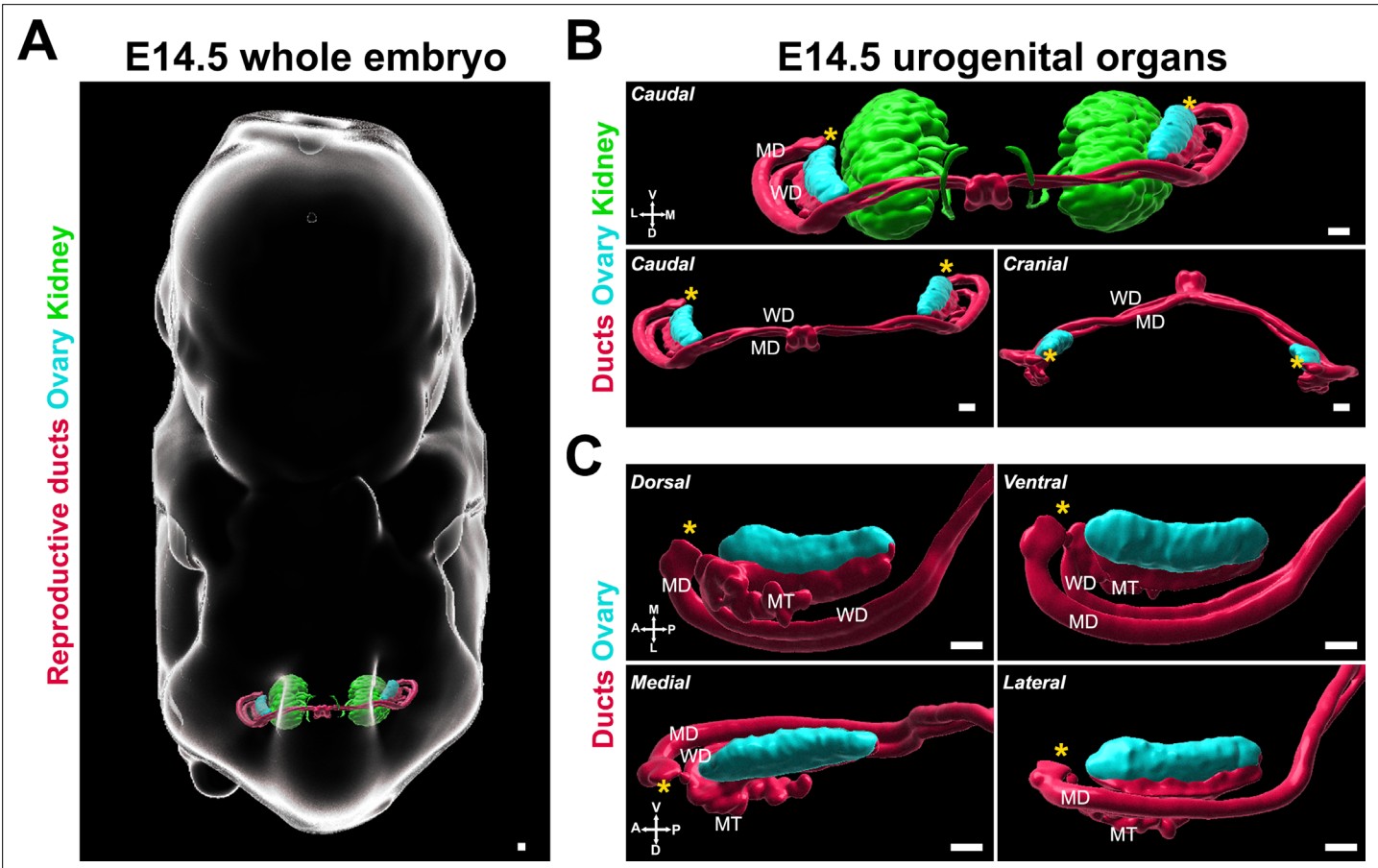

**Figure 1.** 3D models of developing ovaries provide a new perspective of ovary morphology. (**A**) 3D model generated by isosurface segmentation of lightsheet images taken of a whole XX embryo at E14.5 immunostained for PAX8 (kidney, reproductive ducts, *green, red*) and FOXL2 (gonad, *cyan*). (**B**) Caudal and cranial views of the surfaces representing the urogenital system, with and without the kidney surfaces, isolated from the whole embryo images in (**A**). (**C**) Close-ups of dorsal, ventral, medial, and lateral views of the surfaces representing the ovary and reproductive ducts, isolated from the images in (**A**). MD, Müllerian duct; MT, mesonephric tubule; WD, Wolffian duct. Yellow asterisks indicate the location of the infundibulum of the presumptive oviduct for reference. Compasses on the bottom left of each panel indicate the orientation of the ovary: A, anterior; D, dorsal; L, lateral; M, medial; P, posterior; V, ventral. Scale bars, 100 μm.

The online version of this article includes the following video for figure 1:

**Figure 1—video 1.** 3D model of an XX embryo at E14.5.
https://elifesciences.org/articles/81088/figures#fig1video1

the MD is maintained and gives rise to the oviduct (most proximate to the ovary), the uterus, the cervix, and the anterior aspect of the vagina (*Kobayashi and Behringer, 2003*).

We chose E14.5 as the starting stage for our study, as this precedes the major changes in shape and conformation that are specific to the female gonad. To determine the optimal angles to assess ovary morphology, we first performed immunostaining of whole XX embryos at E14.5 with antibodies against FOXL2, which is expressed in pre-granulosa cells and serves as a marker of the gonadal domain, and antibodies against PAX8, which is expressed in the epithelium of the reproductive ducts and cells of the developing kidney. These samples were cleared using iDISCO and imaged with lightsheet microscopy (*Bunce et al., 2021*; *Renier et al., 2014*). We generated 3D models of the labeled structures using isosurface segmentation to provide a morphological view of the urogenital complex in situ (*Figure 1A*, *Figure 1—video 1*). We found that at E14.5, the bilateral gonad/mesonephros complexes resided in the posterior third of the embryo, in a plane orthogonal to the midline axis of the embryo (*Figure 1A*) located on the ventro-lateral aspect of the developing kidneys (*Figure 1B*). A recent study combining contextual 3D images of the developing gonads in the whole embryo with quantitative morphometric analysis showed that the entire genital ridge rotates inward from E10.5 to E12.5 (*Bunce et al., 2021*), thus by E12.5, the gonads appear on the medial aspect of the mesonephros. Consistent with these observations, at E14.5, the reproductive ducts were located lateral to the gonads, which were oriented toward each other in the medial axis (*Figure 1B*). We used the 3D objects representing the ovaries and ducts in this image to define the six views that are presented throughout this study (*Figure 1C*). The caudal and cranial views best illustrated the position and shifts of the ducts relative to the ovary (*Figure 1B*), while the dorsal, medial, lateral, and ventral views provided information on the shape and regionalization of the ovary itself (*Figure 1C*). Dorsal and ventral views also provided context on the position and development of the individual ducts and mesonephric tubules inside the meso-nephros (*Figure 1C*, *top panels*). Lateral and medial views were optimal for observing the dorsal/ventral regionalization of the ovary and the interface between the gonad and the mesonephros (*Figure 1C*, *bottom panels*).

## The ovary transitions from elongated to crescent-shaped during late gestation

Having defined the contextual components of ovary development at E14.5 and the optimal ways of visualizing morphogenetic changes, we next studied the shape of the ovary from E14.5 to birth. For this, we used immunostaining with antibodies against FOXL2 and first imaged whole-mount gonad/mesonephros complexes using confocal microscopy (*Figure 2A*). At E14.5, the ovary was still shaped as an elongated tissue developing on the medial face of the mesonephros. By E17.5, the ovary adopted a rounder conformation. The dorsal side eventually resembled a crescent shape in E18.5 embryos (*Figure 2A*). This change in conformation was accompanied by a hollowing of the FOXL2 domain in the center of the dorsal face of the ovary (*Figure 2A*). We next investigated the dynamics of the FOXL2+ domain in whole ovaries cleared with iDISCO+CUBIC and imaged with lightsheet micros-copy (*Figure 2—figure supplement 1A*; *Figure 2—video 1*). The advantage of this method is that the sample was imaged in its native conformation, in a cylinder of agarose, and thus did not undergo artificial deformation resulting from mounting on a slide (*Figure 2—figure supplement 1A,B*). With these 3D models, we better visualized the conformation of the ovary in space and observed a similar shift from an elongated to a crescent shape between E14.5 and birth (postnatal day 0, P0) (*Figure 2B*). This was most visible on the dorsal and ventral sides of the ovary, where anterior and posterior regions appeared to rise toward the medial axis and come closer to each other. This was associated with an increase in the inflexion toward the center of the ovary, and with an ingression of the medial face of the ovary, leading to an increase in the curvature of the lateral face (*Figure 2B*). Our data thus suggest that the change in conformation of the ovary is representative of a folding process resulting in the center inflexion of the ovary and relocation of anterior and posterior poles.

Upon closer observation, we found that the anterior portion of the ovary appeared to thicken and shift up above the middle of the ovary, suggesting that the anterior region is subject to more deforma-tion than the posterior region (*Figure 2B*). Caudal and cranial views of the process showed that while the ventral surface of the ovary remained smooth during ovary folding, a lateral protrusion extended from the mid-region of the dorsal face between E14.5 and E17.5 (*Figure 2C*, *white arrows*).

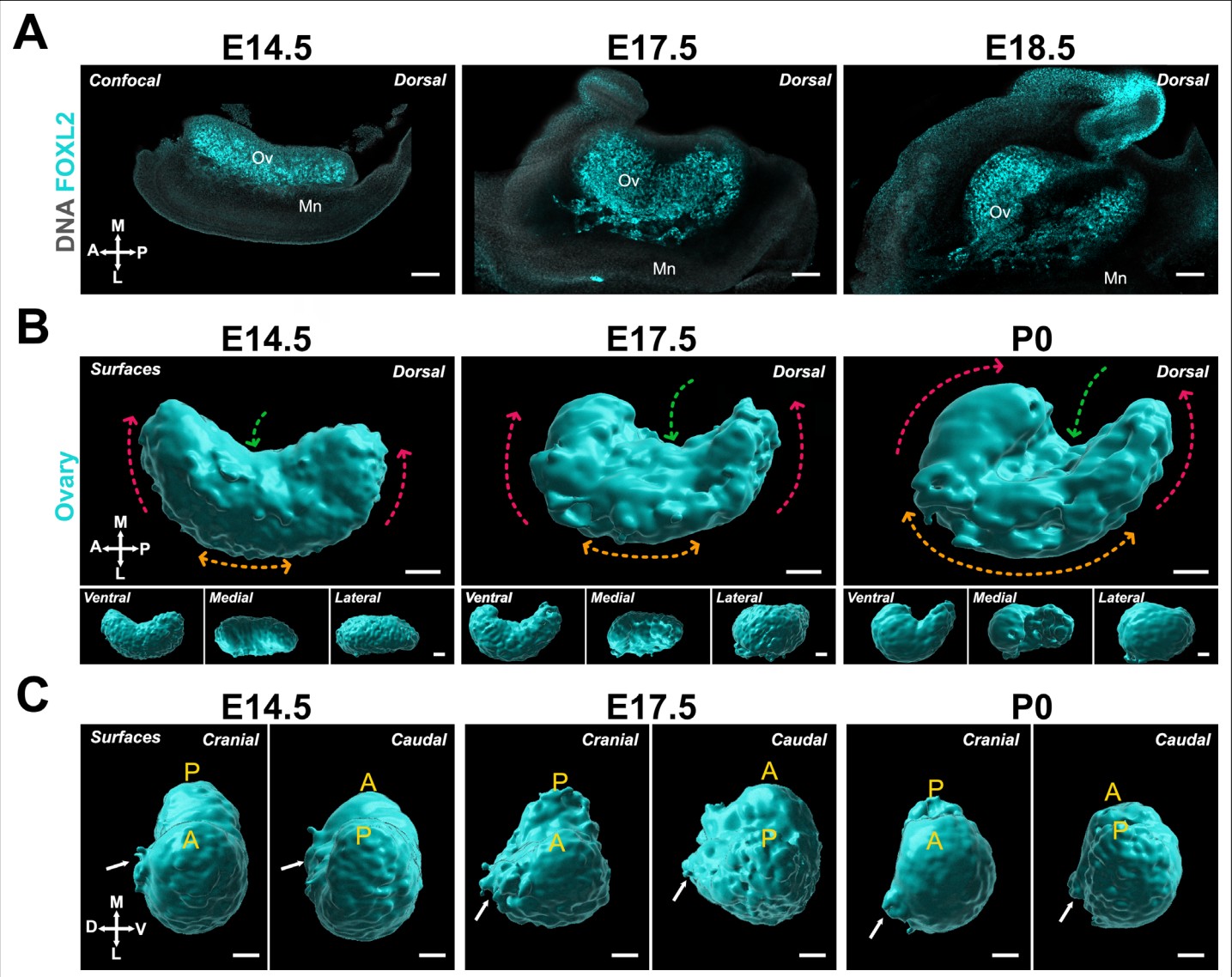

**Figure 2.** The ovary transitions from elongated to crescent-shaped during late gestation. (**A**) Optical sections from confocal Z-stacks of whole immunostaining of ovary/mesonephros complexes at E14.5, E17.5, and E18.5 immunostained for FOXL2 (*cyan*) and counterstained with Hoechst nuclear dye (*grayscale*). (**B**) 3D model generated by isosurface segmentation of lightsheet images taken of whole ovaries at E14.5, E17.5, and P0 immunostained for FOXL2 (*cyan*). Top large panels represent the dorsal view, while small bottom panels illustrate ventral, medial, and lateral views of the same ovary. Dashed arrows illustrate the change in conformation of the ovary as it develops: red arrows, relocation of the anterior and posterior poles; green arrows, medial inflexion; orange arrows, increase in lateral concavity. (**C**) Cranial and caudal views of the surface of the ovaries presented in (**B**). White arrows point to the dorsal protrusion of the ovarian domain. A, anterior pole; MD, Müllerian duct; Mn, mesonephros; Ov, ovary; P, posterior pole. Compasses on the bottom left of each panel indicate the orientation of the ovary: A, anterior; D, dorsal; L, lateral; M, medial; P, posterior; V, ventral. Scale bars, 100 µm.

The online version of this article includes the following video and figure supplement(s) for figure 2:

**Figure supplement 1.** Approach to generate 3D models of the developing ovary.

**Figure 2—video 1.** Imaris workflow.

https://elifesciences.org/articles/81088/figures#fig2video1

## Ovary folding results in specification of the medullary and cortical compartments of the ovary

Previous studies have shown that the location of individual follicles inside the ovary carries functional significance (*Cordeiro et al., 2015*; *Mork et al., 2012*; *Suzuki et al., 2015*; *Zheng et al., 2014*).

Although follicles assembled during late gestation in the medullary compartment of the ovary initiate growth shortly after birth, they become atretic during the juvenile period and are all depleted by the onset of puberty. In parallel, a new wave of de novo follicle formation occurs around birth in the cortical compartment (*Mork et al., 2012*; *Niu and Spradling, 2020*; *Zheng et al., 2014*). These constitute the finite pool of ovarian follicles that remain quiescent throughout the juvenile period and begin to sequentially activate and initiate growth cyclically at the onset of puberty and until menopause, thus supporting the entire female reproductive lifespan (*Duncan and Gerton, 2018*; *Mork et al., 2012*; *Zheng et al., 2014*). While this functional regionalization points to the influence of ovarian architecture on the organization and outcome of follicles within the ovary, both the origin of the cortical and medullary compartments and the nature of the signals and the mechanisms that drive this differential patterning remain unclear. The process of ovary folding as well as the differences in the conformation of the ventral and dorsal surfaces of the ovary led us to investigate whether ovary folding is responsible for the regionalization of the ovary into cortical and medullary compartments. The ovarian cortex is covered with ovarian surface epithelium (OSE), which is the female-specific derivative of the gonadal coelomic epithelium. We used the dynamics of the OSE to serve as a proxy for the establishment of the ovarian cortex during morphogenesis using the *Lgr5-GFP-IRES-CreER* (*Lgr5-Gfp*) transgenic reporter, in which cells of the OSE are labeled with GFP (red in these images) (*Figure 3*; *Ng et al., 2014*). At E14.5, the OSE was only visible in the most medial aspect of the ovary from the dorsal side but covered the ventral portion of the ovarian surface, as demonstrated by the overlap of FOXL2 staining and GFP expression (*Figure 3A and B*). The OSE continued to cover the ventral and lateral faces of the ovary throughout the time-course (*Figure 3C–F*). From E15.5 to P0, the OSE became more visible from the dorsal side as the anterior and posterior domains folded toward the center. As this occurred, the FOXL2+/LGR5− domain became gradually restricted to the dorsal extrusion of the ovarian domain, and to the middle region of the ovary from the dorsal face (*Figure 3C–F*). The appearance of the LGR5+ domain on the dorsal side coincided with the inflexion of the middle region of the ovary, which presumably resulted in an internalization of the cells originally present on the dorsal surface of the ovary to form the ovarian medulla, and to establish the hilum, which serves as a conduit for ovary vasculature and innervation (*Figure 3D, E and F*).

## The expansion and relocation of the MD leave the ovary fully encapsulated

Having characterized the structural changes of the ovarian domain, we next investigated the developmental dynamics of tissues closely associated with the ovary, starting with the MD. The invagination and elongation of the MD occurs from E11.5 to E13.5 in the mouse and is followed by a significant expansion of the duct and functional regionalization into oviduct, uterus, and vagina (*Kobayashi and Behringer, 2003*; *Roly et al., 2018*). The presumptive oviduct develops closest to the ovary and undergoes coiling during late gestation (*Ford et al., 2021*; *Harwalkar et al., 2021*). We investigated the correlations between the dramatic changes in oviduct conformation and ovary morphogenesis. For this, we first used 3D reconstructions of the ovary and MD based on isosurface segmentation of lightsheet images of FOXL2 and PAX8 immunostaining, respectively. The shift in position of the MD with respect to the ovary was best visualized from caudal and cranial views (*Figure 4A*). At E14.5, the MD was fully invaginated and lay parallel to the entire longitudinal axis of the ovary, on its dorso-lateral aspect (*Figure 4A*). The infundibulum is the cranial opening of the presumptive oviduct, which in adults is responsible for capturing ovulated oocytes. The infundibulum was initially located near the anterior pole of the ovary (*Figure 4A*, *yellow asterisk*). At E17.5, while the infundibulum had only slightly shifted to the ventro-medial region of the ovary and the presumptive uterus remained parallel to the longitudinal axis of the ovary, the presumptive oviduct had expanded and shifted ventrally to the ovary (*Figure 4A*). This shift was more pronounced in the anterior region of the oviduct (*Figure 4A*). By P0, the presumptive oviduct had significantly expanded and coiled into its characteristic structure, and this was associated with a complete shift to the ventral side of the ovary (*Figure 4A*).

We next investigated how the relocation of the MD correlated with ovary encapsulation. We immunostained whole-mount gonads with antibodies against PAX8 to label the epithelium of the MD and ENDOMUCIN to label the vasculature (*Brachtendorf et al., 2001*), as a way to visualize the highly vascularized ovarian capsule. These samples were imaged with confocal microscopy (*Figure 4B*). In parallel, we used 3D reconstructions of the ovary, MD, and ovarian capsule based on isosurface

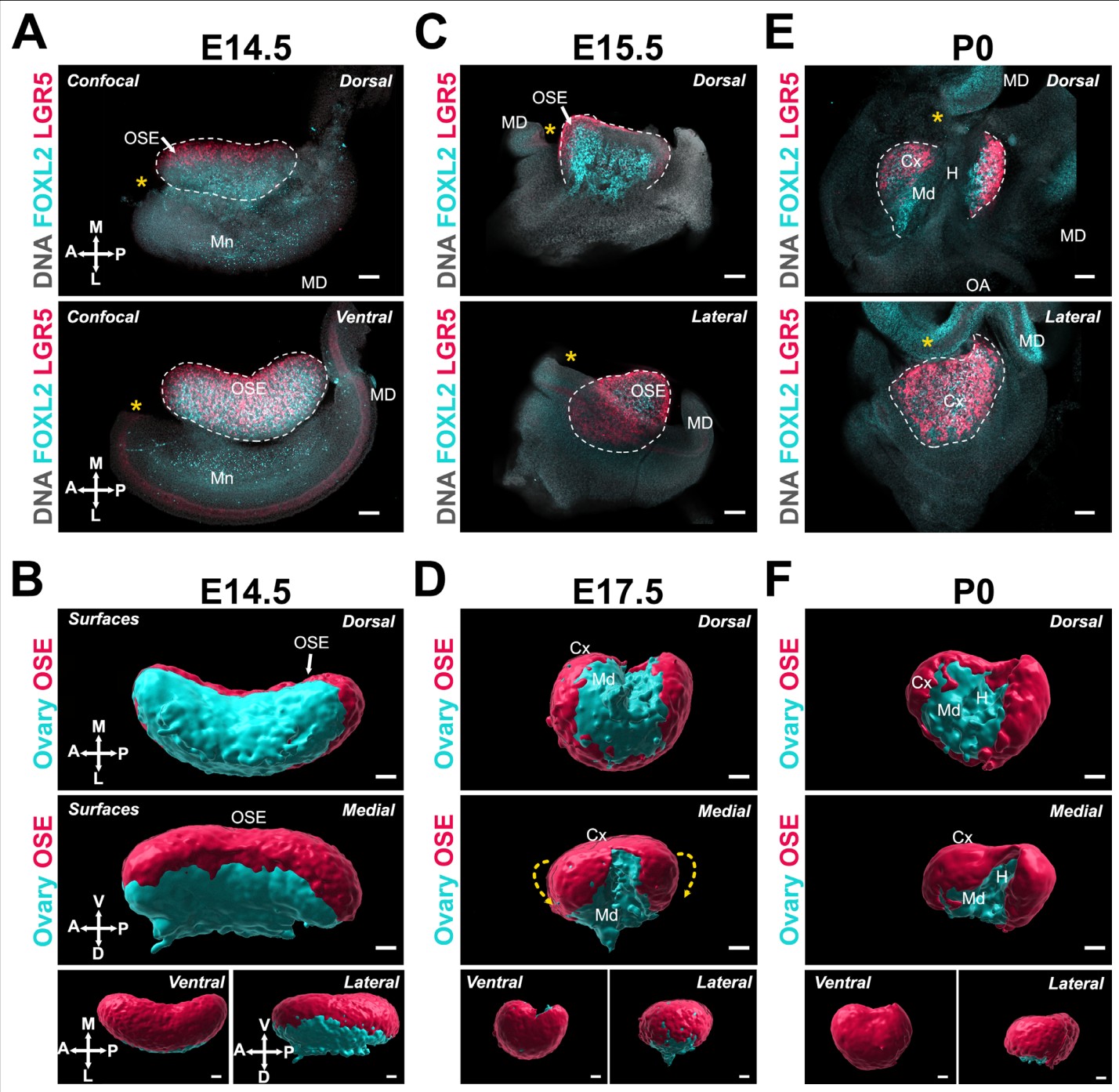

**Figure 3.** Ovary folding is concomitant with specification of the medullary and cortical compartments of the ovary. (**A, C, E**) Optical sections from confocal Z-stacks of whole ovary/mesonephros complexes from *Lgr5-Gfp* mice at E14.5, E15.5, and P0 immunostained for GFP (*red*) and FOXL2 (*cyan*), and counterstained with Hoechst nuclear dye (*grayscale*). Images in the top row were taken from the dorsal side and the bottom row from the ventral (E14.5) or lateral (E17.5; P0). Yellow asterisks indicate the location of the infundibulum of the presumptive oviduct for reference. (**B, D, F**) 3D models generated by isosurface segmentation of lightsheet images taken of whole ovaries from *Lgr5-Gfp* mice at E14.5, E17.5, and P0 immunostained for FOXL2 (*cyan*) and GFP (*red*). Top panels represent the dorsal view, middle panels represent the medial view, and small bottom panels illustrate ventral and lateral views of the same ovary. Yellow dashed arrows in (**D**) indicate the wrapping of the ovarian surface epithelium (OSE) from the ventral to the dorsal side of the ovarian domain. White arrows point to the OSE. Cx, cortex; H, hilum; MD, Müllerian duct; Md, medulla; Mn, mesonephros. Compasses on the bottom left of each panel indicate the orientation of the ovary for the entire row: A, anterior; D, dorsal; L, lateral; M, medial; P, posterior; V, ventral. Scale bars, 100 µm.

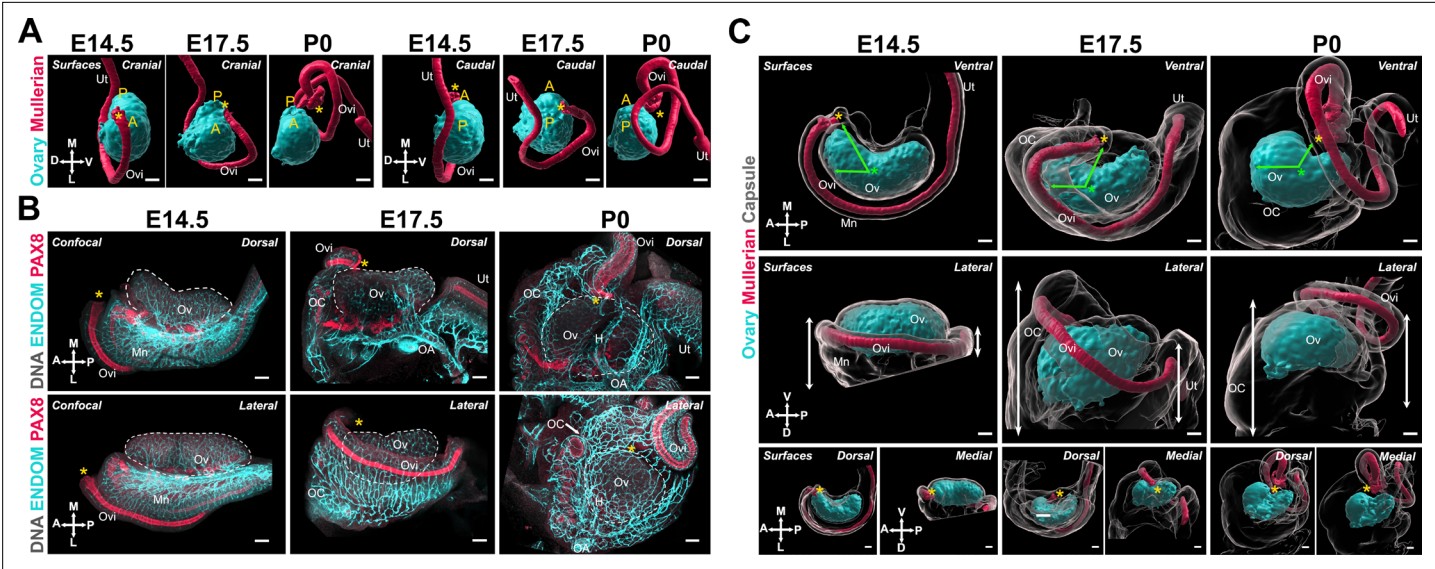

**Figure 4.** The expansion and relocation of the Müllerian duct leave the ovary fully encapsulated. (**A**) 3D model generated by isosurface segmentation of lightsheet images taken of whole ovaries at E14.5, E17.5, and P0 immunostained for FOXL2 (*cyan*) and PAX8 (*red*). A, anterior pole; P, posterior pole. (**B**) Maximum intensity projections from confocal Z-stacks of whole ovary/mesonephros complexes at E14.5 (**B**, *left*), E17.5 (**B**, *middle*), and P0 (**B**, *right*) immunostained for PAX8 (*red*) and ENDOMUCIN (*cyan*), and counterstained with Hoechst nuclear dye (*grayscale*). Images in the top row were taken from the dorsal side and the bottom row from the lateral side. Yellow asterisks indicate the location of the infundibulum of the presumptive oviduct for reference. (**C**) 3D models generated by isosurface segmentation of lightsheet images taken of whole ovaries at E14.5 (**C**, *left*), E17.5 (**C**, *middle*), and P0 (**C**, *right*) immunostained for FOXL2 (*cyan*) and PAX8 (*red*), and counterstained with Hoechst nuclear dye (*grayscale*). The isosurfaces generated with the gray channel allow for visualization of the entire tissue, including the ovarian capsule. White arrowheads point to the Hoechst-based surface of the ovary, which extends beyond the FOXL2+domain, and should not be confused with the ovarian capsule. Top panels represent the ventral view, middle panels represent the lateral view, and small bottom panels illustrate dorsal and medial views of the same ovary. Green arrows indicate the change in position of the infundibulum relative to the longitudinal plane of the ovary. Green asterisks indicate the approximate center of the ovary. White double arrows indicate the extension of mesonephric tissue around the ovary in the anterior (*left arrows*) and posterior regions (*right arrows*). Yellow asterisks indicate the location of the infundibulum of the presumptive oviduct for reference. H, hilum; MD, Müllerian duct; Mn, mesonephros; OA, ovarian artery; OC, ovarian capsule; Ov, ovary; Ovi, oviduct; Ut, uterus. Compasses on the bottom left of each panel indicate the orientation of the ovary for the entire row: A, anterior; D, dorsal; L, lateral; M, medial; P, posterior; V, ventral. Scale bars, 100 μm.

segmentation of lightsheet images of FOXL2, PAX8, and Hoechst labeling, respectively. Consistent with results shown in *Figure 4A*, at E14.5, the MD was located immediately below the ovary, with the infundibulum on the lateral side, near the anterior pole (*Figure 4B*, *left*; *Figure 4C*, *left*). At this stage, the ovary was in direct contact with the abdominal cavity and was not encapsulated. Both the mesonephros and ovary were already highly vascularized (*Figure 4B*, *left*). We used isosurface segmentation of the Hoechst labeling in our lightsheet images to generate surfaces representing the entire ovary/mesonephros complex (*Figure 4C*, *grayscale*) and transparency to make the FOXL2-positive ovarian domain visible (*Figure 4C*, *cyan*). Note that the FOXL2 domain is surrounded by the OSE (not labeled in these images), which accounts for the space between the Hoechst-based isosurface and the FOXL2 domain at all stages (*Figure 4C*, *white double arrows*) and should not be confused with the ovarian capsule, which is absent at E14.5. At E17.5, the infundibulum had progressed from the lateral to the medial portion of the ovary and had begun traveling away from the anterior pole to the center of the ovary (*Figure 4B*, *middle*; *Figure 4C*, *middle*). This was associated with a lateral to medial expansion of cranial mesonephric tissue, the presumptive ovarian capsule, that began covering the cranial region of the ovary (*Figure 4C*, *middle—green arrows*). At this stage, the middle portion of the MD wrapped around the latero-ventral side of the ovary and was no longer visible from the dorsal perspective (*Figure 4B*, *middle*). Thus, the process was best visualized from the lateral view, where the cranial region of the ovary was no longer directly accessible from the abdominal cavity but was instead covered by highly vascularized mesonephric tissue, lined on the medial edge by the expanding MD (*Figure 4B*, *middle*; *Figure 4C*, *middle*). By P0, the infundibulum had continued its cranial to caudal progression and was now located at the middle of the medial side of the ovary, near the opening of the hilum (*Figure 4B*, *right*; *Figure 4C*, *right*). This change was associated with a continued expansion

of cranial mesonephric tissue, which now covered the cranial half of the ovary. At this stage, we also observed a caudal expansion of tissue, and the uterus was now aligned with the infundibulum on the medial side of the ovary (*Figure 4B*, *right*). From the ventral side, the P0 ovary appeared fully encapsulated, with the coils of the oviduct forming the ventral rim of the ovarian capsule (*Figure 4C*, *right*). By the end of this process, the infundibulum of the oviduct was embedded in the capsule, where it later will be responsible for capturing ovulated oocytes.

These data demonstrate that the encapsulation of the ovary occurs during late gestation and is intimately associated with the expansion and regionalization of the MD to form the oviduct. This observation prompted us to investigate potential mechanical components that could be affecting structural change in both the ovary and the oviduct simultaneously.

The broad ligament (BL) and cranial suspensory ligament (CSL) tether the ovary to the MD and body wall and may act as mechanical drivers of ovary folding and encapsulation.

The position of the ovary within the body cavity is maintained by two ligaments: the BL and the CSL. The BL and CSL are peritoneal folds that support the uterus, oviduct, and ovary, securing the position of the reproductive organs inside the body and providing a conduit for vasculature, innervation, and lymphatics (*Martin et al., 1996*). While the BL is a single continuous structure, different segments are defined by the organs they support: the mesometrium supports the uterus, the mesovarium supports the ovary, and the mesosalpinx supports the oviduct. The caudal end of the BL is tethered to the dorsal pelvic body wall, while its cranial region transitions into the CSL, which tethers the ovary to the dorsal abdominal body wall near the diaphragm (*Hutson et al., 1997*; *van der Schoot and Emmen, 1996*). Because these structures are fibromuscular tethers of the ovary, we postulated that developmental dynamics of the BL and CSL could provide insight into the mechanisms of ovary morphogenesis. We first used immunostaining against the neural marker TUJ1 as a proxy for visualizing the BL in whole urogenital complexes from E15.5 to E18.5 (*Figure 5A*). We found that the mesometrium, mesovarium, and mesosalpinx formed a continuous structure on the medial/dorsal aspect of the reproductive tract and ovary at E15.5 and E14.5 (*Figure 5A*, *left*; *Figure 5C*, *left*) labeled also with alpha smooth muscle actin (aSMA) (*Figure 5—figure supplement 1A*, *left*). At E16.5, the mesovarium was visible as a thin sheet of tissue on the medial/dorsal surface of the ovary that hosted neural projections (*Figure 5A*, *middle*) and vasculature (*Figure 5—figure supplement 1A*, *middle*) growing toward the ovary. At E18.5, this sheet of tissue was no longer visible on the medial/dorsal surface of the ovary, and innervation and vasculature had penetrated the dorsal ovary in the region of the hilum (*Figure 5A*, *right*; *Figure 5—figure supplement 1A*, *right*). These observations suggested that the mesovarium, and associated extrinsic components, became internalized into the ovarian medulla as the ovary folded. We next focused on the development of the CSL, which we labeled with antibodies against TNC and aSMA in urogenital complexes from Tg(*Nr5a1-GFP*) mice (*Stallings et al., 2002*), referred to here as SF1-eGFP, in which both the gonad and adrenal are labeled with GFP (*Figure 5B*). We found that at E14.5, the CSL was tightly associated with the initial site of invagination of the MD and extended cranially to the ovary, along the kidney and up to the adrenal (*Figure 5B*, *left*). The adrenal derives from a subpopulation of cells that segregate from gonadal cells early in gonad formation (*Saito et al., 2017*; *Upadhyay and Zamboni, 1982*), suggesting that the TNC+ CSL links these two organs prior to their separation, and throughout development. At P0, the aSMA-labeled CSL remained closely associated with the ovary, kidney, and adrenal (*Figure 5B*, *right*). By this stage, the kidney had significantly grown in volume, resulting in a longer distance between the ovary and adrenal. To investigate the link between the CSL, ovary, and MD in more detail, we labeled isolated gonad/mesonephros complexes at E14.5 and found that the TNC+ domain lined the medial ridge of the ovary and the medial edge of the infundibulum before extending toward the dorsal abdominal wall (*Figure 5C*, *left*). At E18.5, the ovary had folded and the TNC+ CSL continued to line the medial edge of the infundibulum, which had relocated to the opening of the ovarian hilum (*Figure 5C*, *right—yellow asterisk*; *Figure 5—figure supplement 1B*). However, at this stage, the CSL was no longer parallel to the longitudinal axis of the ovary but instead ran through the middle of the ovary before extending toward the dorsal abdominal wall (*Figure 5C*, *right*). These observations were confirmed in semi-transparent 3D models of the ovary and CSL based on lightsheet images, where the ovary was seen folding around the CSL between E14.5 and P0 (*Figure 5—figure supplement 1B*). Thus, rearrangements of the CSL correlated with folding of the ovarian domain and relocation of the infundibulum. Using IF labeling for RUNX1, which is known to label granulosa cells of the ovary and epithelial cells of the MD (*Laronda et al., 2013*;

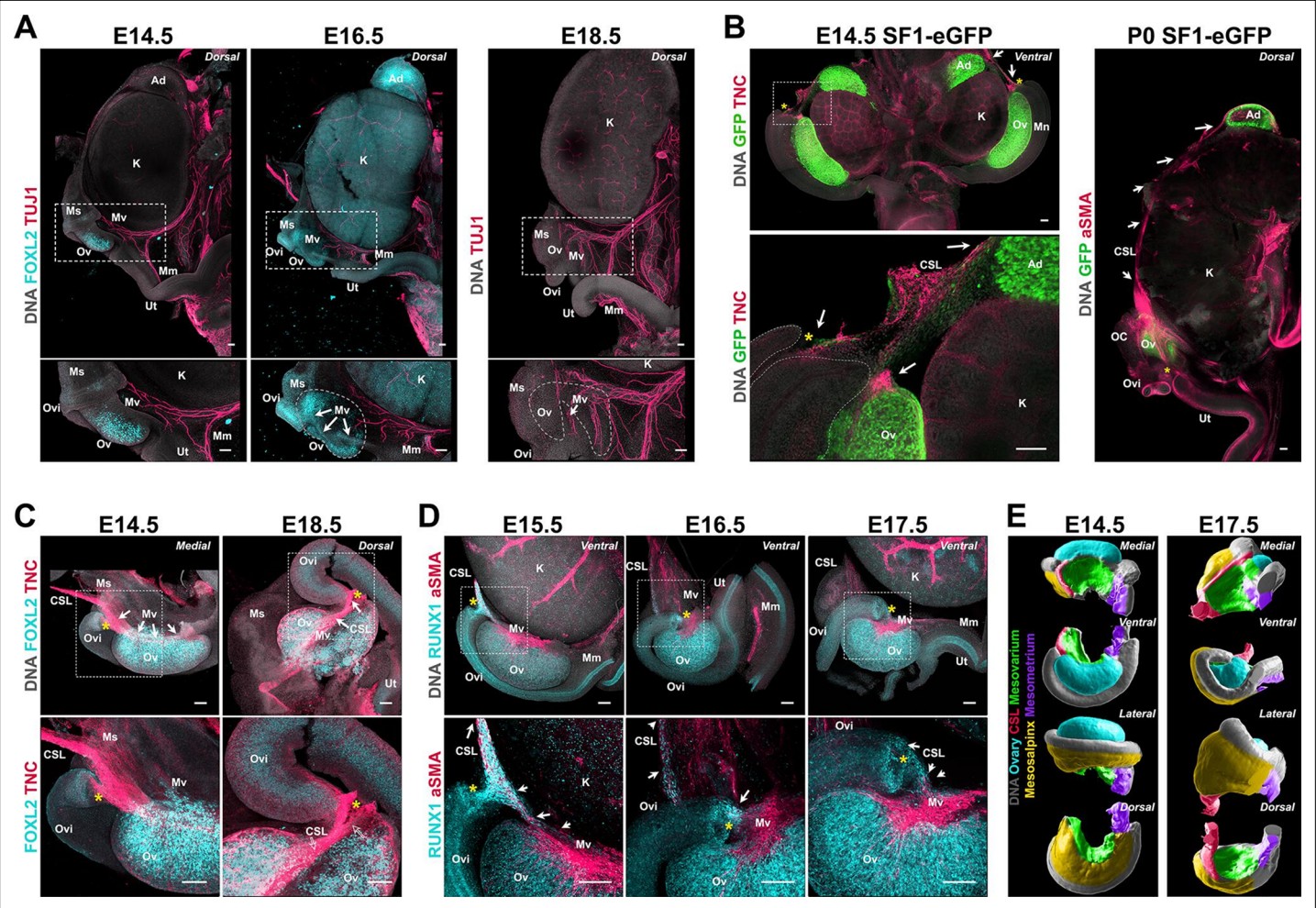

**Figure 5.** Developmental dynamics of the ovarian ligaments correlate with ovary morphogenesis. (**A**) Maximum intensity projections from confocal Z-stacks of whole urogenital complexes at E15.5, E16.5, and E18.5 immunostained for TUJ1 (*red*) and FOXL2 (*cyan—left and middle panel only*), and counterstained with Hoechst nuclear dye (*grayscale*). Images on the bottom row are close-ups of the regions outlined in the top row images. (**B**) Maximum intensity projections from confocal Z-stacks of the whole urogenital complex of XX SF1-eGFP mice at E14.5 (*left*) and P0 (*right*), immunostained for GFP (*green*) and TNC (E14.4, *red*) or aSMA (P0, *red*), and counterstained with Hoechst nuclear dye (*grayscale*). The E14.5 sample was imaged from the ventral side and the P0 from the dorsal. The left bottom panel is a high magnification view of the area outlined in the left top panel. White arrows point to the continuity of the CSL from the ovary to the adrenal. (**C**) Maximum intensity projections from confocal Z-stacks of whole ovary/mesonephros complexes at E14.5 and E18.5 immunostained for FOXL2 (*cyan*) and TNC (*red*), and imaged from the medial side. White arrows point to the continuity of the CSL between the medial ridge of the ovary and the infundibulum. (**D**) Maximum intensity projections from confocal Z-stacks of whole ovary/mesonephros complexes at E15.5, E16.5, and E17.5 immunostained for RUNX1 (*cyan*) and aSMA (*red*), and imaged from the ventral side. White arrows point to the population of RUNX1+ cells linking the medial edge of the mesovarium to the infundibulum, and to the CSL. (**E**) 3D models generated by isosurface segmentation of lightsheet images taken of whole ovary/mesonephros complexes at E14.5 (*left*) and E17.5 (*right*) stained with Hoechst nuclear dye (*grayscale*), and false-colored to illustrate the three regions of the BL (*purple*, mesometrium; *yellow*, mesosalpinx; green, mesovarium), the CSL (*red*), and the ovary (*cyan*). Each row represents a different view of the same ovary for each stage (from top to bottom: *medial*; *ventral*; *lateral*; *dorsal*). Yellow asterisks indicate the location of the infundibulum of the presumptive oviduct for reference. Ad, adrenal; CSL, cranial suspensory ligament; K, kidney; Mm, mesometrium; Mn, mesonephros; Ms, mesosalpinx; Mv, mesovarium; OC, ovarian capsule; Ov, ovary; Ovi, oviduct; Ut, uterus. Scale bars, 100 μm.

The online version of this article includes the following figure supplement(s) for figure 5:

**Figure supplement 1.** The mesovarium and cranial suspensory ligament tether the ovary to the rest of the urogenital complex.

*Nicol et al., 2019*), we discovered that RUNX1 also labels a discrete population of cells present at the medial edge of the aSMA+ mesovarium, which appeared to form a continuous structure from the mesovarium to the infundibulum to the body wall from E14.5 to E17.5 (*Figure 5D*). These data suggest that the CSL acts as a physical tether that links the ovary to the infundibulum and to the body

wall throughout development. Therefore, any mechanical tension on the CSL, caused for example by the growth of the kidney, could direct the relocation of the infundibulum and the folding of the ovarian domain simultaneously. The false-colored 3D models in *Figure 5E* represent all four views of ovary/mesonephros complexes at E14.5 and E17.5, and clearly illustrate the expansion of the mesosalpinx that correlates with translocation of the MD, and results in ovary encapsulation (*Figure 5E*, *yellow*) and internalization of the mesovarium during ovary folding (*Figure 5E*, *green*).

## Developmental dynamics of the RO are integrated with events of ovary morphogenesis

The RO is an epithelial appendage of the ovary thought to arise from the mesonephric tubules (*Satoh, 1985*). The RO was first described in 1870 by W. Waldeyer (as cited by *Byskov and Lintern-Moore, 1973*), and was believed to be a functionless female homolog of the male rete testis (*Kulibin and Malolina, 2020*; *Wenzel and Odend'hal, 1985*). It was once thought to degenerate in most individuals, however research shows that this structure persists into adulthood in large mammals (*Archbald et al., 1971*; *Hadek, 1958*; *Mossman and Duke, 1975*). The RO consists of three distinct regions: (1) the intraovarian rete (IOR), (2) the connecting rete (CR), and (3) the extraovarian rete (EOR) (*Byskov, 1978*). The IOR resides inside of the ovary and consists of thin strings of squamous cells. The EOR is a tubular epithelial structure that resides outside of the ovary, and the CR is a transitional zone between the two regions (*Byskov and Lintern-Moore, 1973*; *Lee et al., 2011*). Previous literature has not explored the development of the RO or a possible role during morphogenesis of the ovary.

As the RO is directly linked to the ovary, we hypothesized that morphogenesis of the RO might be involved in morphogenesis of the ovary. To investigate the morphological changes of the RO during development, we used antibodies against PAX8, which labels all three RO regions, to image the ovary and RO using confocal and lightsheet microscopy. Our findings demonstrated that the regions of the RO changed in proportion to one another during the embryonic and early postnatal period. At E14.5, the RO was exclusively located on the dorsal side of the ovary and all three regions were distinguishable (*Figure 6A*, *E14.5*; *Figure 6B*, *left*). The IOR was the largest of the three regions at this stage and lined most of the interface between the ovary and mesonephros. PAX8+ cells were squamous and intermingled with FOXL2+ cells, suggesting the IOR was embedded inside the ovarian domain. The rest of the structure was connected to the IOR and located within the mesonephric tissue. The region proximate to the ovary was characterized by a short region of transition from squamous epithelial cells in the CR to a columnar epithelium in the EOR, which extended in tubules outward from the ovary (*Figure 6A*, *E14.5*; *Figure 6B*, *left*). As the ovary folded between E16.5 and E17.5, the IOR adopted a U-shape (*Figure 6A*, *E16.5–E17.5*; *Figure 6B*, *middle*). At E16.5, the tubules of the EOR began to elongate and by E17.5 had wrapped around to the ventral face of the ovary (*Figure 6A*, *E16.5–E17.5*; *Figure 6B*, *E17.5*). The CR was located alongside the dorsal protrusion of the ovary of this stage (*Figure 6A*, *E16.5*; *Figure 6B*, *middle*). At E17.5, the embedded IOR was restricted to the edge of the FOXL2 domain on the dorsal side (*Figure 6A*, *E17.5*), and the CR was more predominant with growth of the dorsal protrusion. The EOR underwent rapid growth and convolution of the tubules, especially apparent in the 3D model, where both the dorsal and medial views demonstrated the intricacy of the tubular coiling (*Figure 6B*, *right*). By E18.5, the IOR had slightly regressed compared to E17.5, and no longer extended all the way to the poles of the ovarian domain (*Figure 6A*, *E18.5*). At this stage, the EOR was the largest of the three regions, and had developed into an elaborate and convoluted tubular structure that curved alongside the ovary within the ovarian capsule (*Figure 6A*, *E18.5*). Between P0 and P5, the IOR mostly regressed to the lateral edge of the medulla, but a thin string of cells continued to line the dorsal boundary between the ovarian cortex and medulla, and the EOR continued to grow and wrap around the ovary (*Figure 6A*, P0–P5; *Figure 6B*, *right*). The location of the IOR at the boundary of the ovarian cortex coincides with the dorsal opening of the hilum. To further characterize this domain, we stained late-gestation ovaries for ENDOMUCIN and TUJ1, which label vasculature and innervation, respectively. These components entered the ovary through this dorsal opening, lined with cells of the IOR (*Figure 6—figure supplement 1A, B*). This suggests that the IOR plays a role in the invasion of extrinsic components into the ovarian medulla, either by mechanically restricting folding on the dorsal side, by producing chemoattractant signals, or both.

The morphological difference between cells of the EOR and CR/IOR prompted us to investigate the expression of tubular epithelium markers KRT8 and E-Cadherin. We found that these

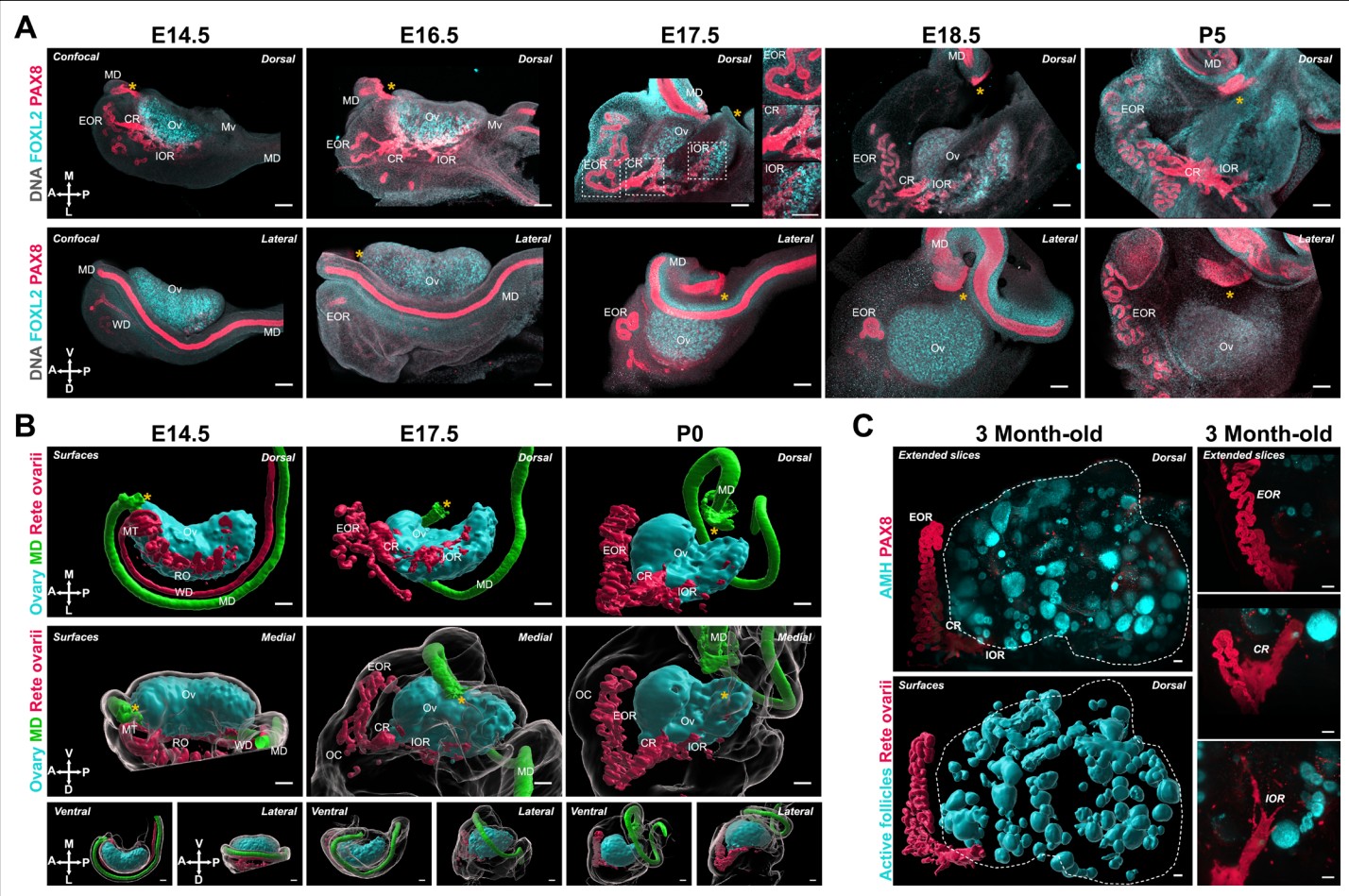

**Figure 6.** Developmental dynamics of the rete ovarii correlate with ovary morphogenesis. (**A**) Maximum intensity projections from confocal Z-stacks of whole ovary/mesonephros complexes at E14.5, E16.5, E17.5, E18.5, and P5, immunostained for FOXL2 (*cyan*) and PAX8 (*red*), and counterstained with Hoechst nuclear dye (*grayscale*). Samples in the top row were imaged from the dorsal side and samples from the bottom row from the lateral side. Side panels in (**A**, *E17.5*) are close-ups of the three regions of the RO outlined at E17.5 (from top to bottom: EOR, CR, and IOR). (**B**) 3D models generated by isosurface segmentation of lightsheet images taken of whole ovaries at E14.5, E17.5, and P0 immunostained for FOXL2 (*cyan*) and PAX8 (Müllerian duct, *green*; and Rete ovarii, *red*). All samples were counterstained with Hoechst nuclear dye (*grayscale*). Top panels represent the dorsal view, middle panels represent the medial view, and small bottom panels illustrate ventral and lateral views of the same ovary. Yellow asterisks indicate the location of the infundibulum of the presumptive oviduct for reference. Compasses on the bottom left of each panel indicate the orientation of the ovary: ; D, dorsal; L, lateral; M, medial; V, ventral. (**C**) Lightsheet images of whole ovaries from 3-month-old mice immunostained for AMH (*cyan*) and PAX8 (*red*). The top panel shows the extended slice view of the native image, while the bottom panel illustrates the 3D surfaces of the RO (PAX8, *red*) and ovarian follicles (AMH, *cyan*) generated by isosurface segmentation of the same image. Images in the right panel are close-ups of the three regions of the RO in an adult ovary: top—EOR; middle—CR; bottom—IOR. CR, connecting rete; EOR, extraovarian rete; IOR, intraovarian rete; MD, Müllerian duct; Mn, mesonephros; MT, mesonephric tubules; Mv, mesovarium; OC, ovarian capsule; Ov, ovary; RO, rete ovarii; WD, Wolffian duct. Scale bars, 100 μm.

The online version of this article includes the following figure supplement(s) for figure 6:

**Figure supplement 1.** The IOR and CR coincide with the point of entry of vasculature and innervation into the ovary.

**Figure supplement 2.** Expression of known epithelial and gonadal markers in the different regions of the rete ovarii.

markers were significantly enriched in the EOR (*Figure 6—figure supplement 2A, B*). The EOR was also the only region to be wrapped in a aSMA+ sheath, presumably fibromuscular tissue of mesenchymal origin (*Figure 6—figure supplement 2C*). We also used antibodies against SOX9, a broader epithelial marker, and noted that all three regions of the RO were labeled (*Figure 6—figure supplement 2D*). A small subset of the CR and IOR cells were positive for the pre-granulosa marker FOXL2, suggestive of a transitional region. The CR and IOR also expressed several ovarian markers, including RUNX1, SF-1, and GATA4, reinforcing the notion that cells of the IOR are closely related to ovarian cells (*Figure 6—figure supplement 2E,F,G*). This is consistent with a recent single-cell

transcriptomic study of mouse gonad development showing that cells of the RO were closely related to pre-supporting cells, and even gave rise to a subset of pre-granulosa cells during early differentiation of the ovary *Mayère et al., 2022*. While expression of RUNX1 was lower in the CR, this was the only gonadal marker to be expressed in the tubular cells of the EOR (*Figure 6—figure supplement 2E*). In addition, we found that some cells of the IOR and CR expressed both PAX8 and the gonadal markers *Nr5a1* (SF1-eGFP) and FOXL2 (*Figure 6—figure supplement 2F, H*, *white arrows*). Interestingly, PAX8+ cells of the IOR were observed in close contact with oocytes labeled with HuC/D, suggesting that the IOR was intermingled with ovarian follicles in the medullary region (*Figure 6—figure supplement 2H*). We next investigated the presence and structure of the RO in adult mouse ovaries using tissue clearing and lightsheet microscopy. The RO was labeled with PAX8 antibodies, and the ovarian domain was identified using antibodies against AMH (Anti-Müllerian Hormone—a marker of preantral and antral follicles). All three regions of the RO persisted into adulthood (*Figure 6C*), suggesting that the role of the RO is not limited to the embryonic period. In adulthood, the EOR remained the largest of the regions followed by the CR and IOR (*Figure 6C*). Of note, Proteintech's 10336-1-AP polyclonal rabbit anti-PAX8 antibody has cross-reactivities with other PAX family members (*Moretti et al., 2012*), thus a detailed expression analysis for different Pax family genes in the RO may provide more precise information on the expression and putative role of these genes in the RO.

## Combinations of Pax2 and Pax8 deletion alleles differentially affect the oviduct and regions of the rete ovarii

*Pax2* and *Pax8* are essential genes during the development of the urogenital system, and null alleles for both of these have been shown to impact the development of the kidneys and the reproductive ducts in mice (*Torres et al., 1995*; *Mittag et al., 2007*). While Pax8 is expressed in all three regions of the RO, Pax2 is not expressed in the intraovarian progenitors of the RO (*Mayère et al., 2022*). Both genes are expressed in cells of the oviduct throughout development (*Torres et al., 1995*). Thus, to directly test the role of proximate tissues on ovary morphogenesis, we established a genetic model for ubiquitous deletion of *Pax2* and *Pax8*. We first generated an allelic series and immunostained ovary/oviduct complexes with antibodies against KRT8 and PAX8 to study the impact of loss of *Pax2*, *Pax8*, or both, on the oviduct and the RO at E18.5 (*Figure 7A*). The oviduct appeared unaffected by the absence of one copy of either *Pax2* or *Pax8* (*Figure 7A*, *top row-second and third columns*). While deletion of a single copy of *Pax2* (*Pax2^{del/+}*; *Pax8^{+/+}*) did not appear to impact the oviduct or any region of the RO (*Figure 7A*, *middle and bottom row-second column*), deletion of a single copy of *Pax8* (*Pax2^{+/+}*; *Pax8^{del/+}*) led to a decrease in the size of the IOR and the CR, which was no longer connected to the EOR (*Figure 7A*, *bottom row-third column*). When one copy of *Pax2* and one copy of *Pax8* were deleted (*Pax2^{del/+}*; *Pax8^{del/+}*), the MD began to display perturbations, such as gaps in the epithelium or a blunt end near the uterus (*Figure 7A*, *fourth column*, *white arrows*). In addition, all three regions of the RO were affected: the IOR and CR were diminished similarly to *Pax2^{+/+}*; *Pax8^{del/+}* samples, and the EOR was sparse with less convoluted tubules (*Figure 7A*, *fourth column*, *middle and bottom rows*). Homozygous deletion of *Pax2* (*Pax2^{del/del}*; *Pax8^{+/+}*) or *Pax8* (*Pax2^{+/+}*; *Pax8^{del/del}*) severely affected the oviduct, which displayed large gaps and blunt distal ends. However, in both cases, a significant portion of the oviduct proximal to the ovary was continuous (*Figure 7A*, *top row-fifth and sixth column*). While the IOR and CR persisted in animals with homozygous deletion of *Pax2* (*Pax2^{del/del}*; *Pax8^{+/+}*), the EOR was absent. In contrast, all three regions of the RO were affected by homozygous deletion of *Pax8*: the IOR and CR were completely absent, while the EOR appeared fragmented (*Figure 7A*, *middle and bottom row-sixth column*). To summarize, this allelic series shows that development of a continuous oviduct and EOR requires two functional copies of either Pax2 or Pax8 plus one functional copy of the other (*Supplementary file 1a*). The intraovarian and connecting regions of the RO do not require expression of *Pax2*, nor does *Pax2* compensate for loss of *Pax8* in these regions (*Supplementary file 1a*). Both copies of *Pax8* are required to form the IOR, while one copy is sufficient for the development of the CR (*Supplementary file 1a*). Importantly, all ovaries from this allelic series were encapsulated and ovary folding was not obviously affected, suggesting that these morphogenetic processes do not rely on the presence of a continuous oviduct or the RO at E17.5.

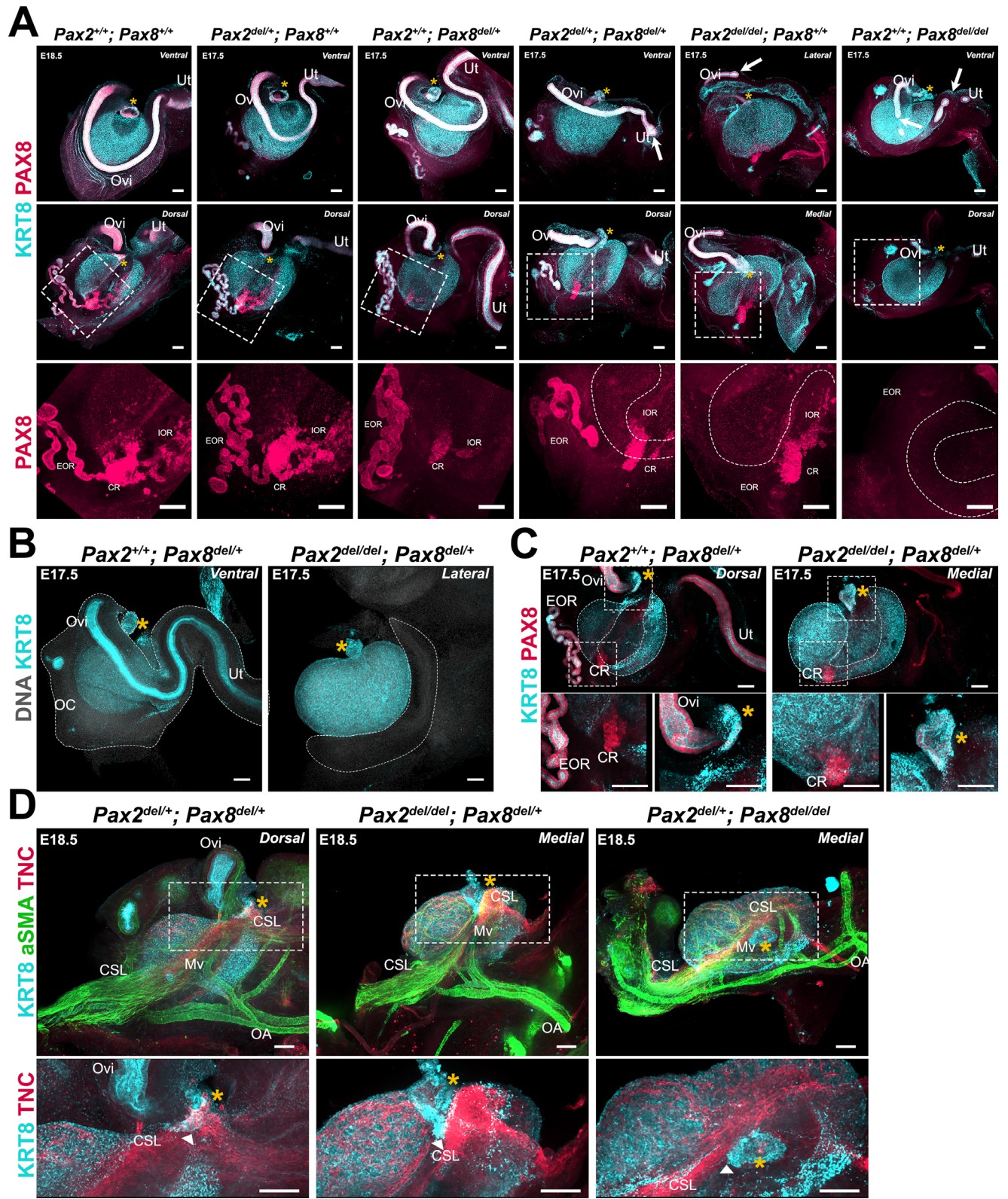

**Figure 7.** Perturbations of the Müllerian duct (MD) and rete ovarii by deletion of Pax2 and/or Pax8 disrupt ovary morphogenesis. (**A**) *Pax2/Pax8* deletion allelic series. Maximum intensity projections from confocal Z-stacks of whole ovary/mesonephros complexes at E17.5 and E18.5, immunostained for KRT8 (*cyan*) and PAX8 (*red*). Each sample is representative of the phenotype observed for different combinations of Pax2 and Pax8 deletion alleles. Images in the top row were captured from the dorsal side, and images in the middle, and bottom row from the dorsal side. Images in the bottom row

*Figure 7 continued on next page*

*Figure 7 continued*

are close-ups of the regions outlined in the middle row. White arrows point to disruptions of the MD. Dashed lines in the bottom row represent the KRT8+ cortical domain of the ovary. (B) The segment of the oviduct is required for ovary encapsulation. Maximum intensity projections from confocal Z-stacks of whole ovary/mesonephros complexes at E17.5, immunostained for KRT8 (*cyan*) and counterstained with Hoechst nuclear dye (*grayscale*). The dashed outline represents the ovarian capsule, which is absent in *Pax2$^{del/del}$*; *Pax8$^{del/+}$* samples. (C) Only the infundibulum and connecting rete persist in *Pax2$^{del/del}$*; *Pax8$^{del/+}$* samples. Maximum intensity projections from confocal Z-stacks of whole ovary/mesonephros complexes at E17.5, immunostained for KRT8 (*cyan*) and PAX8 (*red*). Images in the bottom row are close-ups of the regions outlined in the top row: left, rete ovarii; right, infundibulum. (D) The cranial suspensory ligament (CSL) remains tethered to the infundibulum remnant in *Pax2$^{del/del}$*; *Pax8$^{del/+}$* and *Pax2$^{del/+}$*; *Pax8$^{del/del}$* samples. Maximum intensity projections from confocal Z-stacks of whole ovary/mesonephros complexes at E18.5, immunostained for KRT8 (*cyan*), aSMA (*green*), and TNC (*red*). Images in the bottom row are close-ups of the regions outlined in the top row. White arrowheads point to the attachment of the CSL to the infundibulum. Yellow asterisks indicate the location of the infundibulum of the presumptive oviduct for reference. CR, connecting rete; EOR, extraovarian rete; IOR, intraovarian rete; Mv, mesovarium; OA, ovarian artery; OC, ovarian capsule; Ov, ovary; Ovi, oviduct, Ut, uterus. Scale bars, 100 μm.

The online version of this article includes the following figure supplement(s) for figure 7:

**Figure supplement 1.** Gross morphology of *Pax2$^{del}$*; *Pax8$^{del}$* fetuses.

## While a segment of oviduct is required for ovary encapsulation, the infundibulum is sufficient to tether the ovary to the cranial suspensory ligament

We next examined the development of the RO and MD in *Pax2$^{del/del}$*; *Pax8$^{del/+}$* fetuses. We first noted that *Pax2$^{del/del}$*; *Pax8$^{del/+}$* and *Pax2$^{del/+}$*; *Pax8$^{del/del}$* presented with kidney agenesis (*Figure 7—figure supplement 1*), consistent with the phenotypes previously observed using combinations of null alleles for *Pax2* and *Pax8* (*Bouchard et al., 2002*; *Laszczyk et al., 2020*; *Torres et al., 1995*). We found in these samples that the ovaries were located higher in the body cavity, much closer to the adrenal than in controls (*Figure 7—figure supplement 1*). This suggests that growth of the kidney pushes the ovary to a more caudal position during development. Using antibodies against KRT8 to label the Müllerian epithelium, we found that *Pax2$^{del/del}$*; *Pax8$^{del/+}$* mice lacked most of the MD, including the segment of the oviduct that typically wraps around the ovary (*Figure 7B*, *Supplementary file 1a*). This was associated with absence of the ovarian capsule, demonstrating that a continuous oviduct is required for growth of the mesosalpinx that leads to ovary encapsulation (*Figure 7B*, *Supplementary file 1a*). We next examined the expression of PAX8 and KRT8 on the dorsal side of *Pax2$^{del/del}$*; *Pax8$^{del/+}$* ovaries compared to single deletion of *Pax8* (*Pax8$^{del/+}$*), and found that the IOR and EOR were absent, while a fragment of the CR persisted (*Figure 7C*, *Supplementary file 1a*). In addition, we observed a small fragment of KRT8+/PAX8+ tissue located at the medial opening of the folded ovary (*Figure 7C*, *top and bottom right panels*). The location and immunolabeling indicated that this was the infundibulum of the oviduct, which persisted despite the deletion of both copies of *Pax2* and one copy of *Pax8*. Interestingly, the position of the infundibulum in these mice was similar to that in controls, suggesting that the location of the infundibulum at the medial edge of the ovary at E17.5 does not rely on the presence of a continuous oviduct (*Figure 7C*). These ovaries also appeared to have folded. This observation was consistent with our hypothesis that the infundibulum is tethered to the ovary through the CSL, and that this bond may be responsible for both ovary folding and relocation of the infundibulum. To test whether the CSL remained attached to the ovary and to this small fragment of infundibulum, we labeled the mesovarium and CSL of *Pax2$^{del/del}$*; *Pax8$^{del/+}$* and *Pax2$^{del/+}$*; *Pax8$^{del/del}$* samples using antibodies against aSMA and TNC (*Figure 7D*). In both genotypes, the CSL was present, and the mesovarium appeared to become internalized on the dorsal side as the ovary folded, similarly to *Pax2$^{del/+}$*; *Pax8$^{del/+}$* controls, in which a longer segment of oviduct was present (*Figure 7D*, *top panels*). Importantly, the TNC+ CSL was present and appeared tethered to the medial edge of the ovary and to the persistent fragment of infundibulum in both *Pax2$^{del/del}$*; *Pax8$^{del/+}$* and *Pax2$^{del/+}$*; *Pax8$^{del/del}$* conditions (*Figure 7D*, *bottom panels*). Overall, these data suggest that while the oviduct is required for ovary encapsulation, the infundibulum is sufficient to tether the ovary to the CSL.

## Discussion

While the cortical/medullary structure of the mature ovary is well characterized, the mechanisms of organogenesis underlying the transition from the elongated structure of the early gonad had not

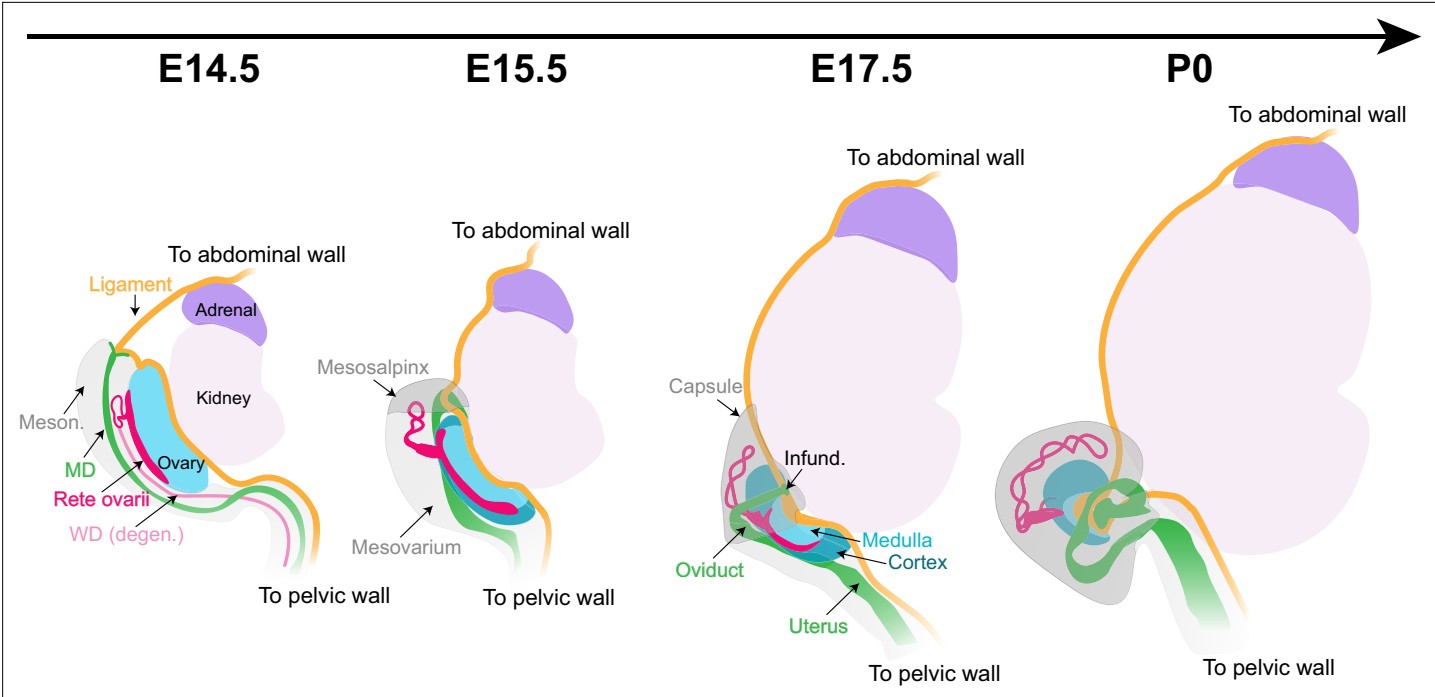

**Figure 8.** Morphogenesis of the fetal mouse ovary. The dorsal face of the ovary is the center of action for most architectural changes during ovary morphogenesis. The cranial suspensory ligament (CSL, *yellow*), which tethers the ovary to the dorsal body wall, originally appears on the medial ridge of the ovary (*blue*). Throughout ovary morphogenesis, the cranial region of the oviduct (*green*), the infundibulum, remains linked to the original anterior pole of the ovary through their mutual attachment to the ligament. Ovary folding is associated with relocation of the infundibulum to the medio-ventral opening of the ovary. The relocation of the oviduct and infundibulum is driven by the dorsal expansion of mesonephric tissue (*gray*) to form the mesosalpinx and mesovarium. This process leaves the ovary fully encapsulated, with the ovarian capsule arising from the growth and fusion of the mesovarium and the mesosalpinx (*gray*). The developmental dynamics of the rete ovarii (*red*) follow the morphogenesis of the ovary, with the intraovarian and connecting regions closely associated with the ovarian domain throughout folding, and the extraovarian rete expanding within the developing ovarian capsule. Infund., infundibulum; MD, Müllerian duct; Meson., mesonephros; WD, Wolffian duct.

been explained. The location of individual follicles within the cortex or medulla of the ovary appears to be crucial in determining their outcome (*Cordeiro et al., 2015*; *Mork et al., 2012*; *Suzuki et al., 2015*), but the mechanisms that control regionalization of the ovary remain poorly understood. Using lightsheet and confocal imaging, we provide a robust image data set that catalogs the complex events of ovary morphogenesis in the context of the extrinsic structures incorporated into the ovary and the other female reproductive structures that surround it. Importantly, the morphogenetic data presented here are an important first step toward integrating mechanical and architectural aspects of ovary development with the establishment of the medullary and cortical domains of the ovary, and the establishment of female fertility. In the present study, we propose a model of ovary morphogenesis, illustrated in *Figure 8*, that links the establishment of ovarian architecture and the functional regionalization of the ovary into cortical and medullary compartments with the morphogenesis of the oviduct and the CSL.

The dorsal face of the ovary is the center of action for most architectural changes during ovary morphogenesis. While the ovary initially develops on top of the mesonephros, the mesovarium progressively lines the dorsal aspect of the ovary. The CSL, which tethers the ovary to the dorsal body wall, originally appears on the medial ridge of the ovary, parallel to its longitudinal axis (*Figure 8*). Throughout ovary morphogenesis, the cranial region of the oviduct, the infundibulum, remains linked to the original anterior pole of the ovary through their mutual attachment to the CSL (*Figure 8*). Ovary folding is associated with the relocation of the infundibulum to the medio-ventral opening of the ovarian hilum and incorporation of the CSL within the fold (*Figure 8*). The relocation of the oviduct and infundibulum is driven by the dorsal expansion of the mesovarium and the cranial expansion of the mesosalpinx, the region of the BL that supports the oviduct (*Harwalkar et al., 2021*). This process leaves the ovary fully encapsulated, with the ovarian capsule arising from the growth and fusion of

the mesovarium and the mesosalpinx (*Agduhr, 1927*; *Figure 8*). Using *Pax2* and *Pax8* deletion alleles as a functional model to investigate the role of the oviduct during ovary morphogenesis, we found that a continuous oviduct is required for ovary encapsulation. It will be interesting in the future to use this model to investigate whether the isolation of the ovary afforded by the capsule is significant for regionalization of the ovary into cortical and medullary compartments. We hypothesize that mechanical tension on the CSL drives the process of ovary folding. As the ovary folds toward its dorsal face, the mesovarium and associated vasculature and innervation, initially located on the medio-dorsal aspect of the ovary, become internalized inside the folded ovary, contributing new tissue to the initial gonad primordium. As the ovary folds, the OSE, located as a canopy on the ventral face of the ovary at E14.5, expands to the dorsal face. This process establishes the ovarian cortex and medulla (*Figure 8*). Oocytes and pre-granulosa cells that were originally in the dorsal region of the ovary become located in the medulla, in close proximity to newly internalized extrinsic components. This suggests that the allocation of follicles into medullary and cortical domains is driven by structural changes of the ovary during morphogenesis. It will be an exciting endeavor to integrate the morphogenetic events presented here with the dynamics of ovarian follicle formation to decipher the contribution of extrinsic molecular and mechanical signals to the establishment of the ovarian reserve.

We initially hypothesized that growth of the kidney induces tension on the CSL to drive ovary folding and relocation of the infundibulum. However, in *Pax2*$^{del/del}$; *Pax8*$^{del/+}$ and *Pax2*$^{del/+}$; *Pax8*$^{del/del}$ mice, which retain the CSL and lack kidneys, the ovary folded to varying degrees, suggesting that growth of the kidney is not the sole driver of ovary folding. Our proposed model for ovary morphogenesis points to the CSL as the driver of the cascade of events that lead to the establishment of the architecture of the ovary. A previous study demonstrated that the gut mesentery, which is a fibromuscular connective tissue similar to the BL and CSL, plays a crucial role in the establishment and maintenance of gut looping by exerting contractile forces on the developing intestine (*Savin et al., 2011*). One possibility is that the CSL displays autonomous contractility. It is also possible that several adjacent tissues contribute to ovary folding, including the kidney, CSL, infundibulum, and RO. Additional experiments will be necessary to tease apart the mechanical and molecular aspects of ovary folding. While the CSL originally develops in both males and females, it is maintained only in females to secure the ovary to the abdominal wall. In males, the CSL regresses to accommodate testis descent (*Adham et al., 2002*; *Hutson et al., 1997*; *van der Schoot and Emmen, 1996*). Although the testis does not normally undergo folding, it will be interesting in the future to determine whether some folding of the testis occurs in Persistent Mullerian Duct syndrome. In addition, while the testis is not permissive to the invasion of innervation (*McKey et al., 2019*; *Svingen et al., 2012*), the ovary integrates extrinsic components such as neurons and vasculature within the medulla, at least in part, as a result of folding over the growing mesovarium. The integration of these components likely influences the functional regionalization of the ovary.

This study also illuminates the developmental dynamics of the RO (*Figure 8*). We found that the CR remains at the interface between the ovary and mesonephric tissue throughout development and corresponds to the dorsal protrusion of the ovarian domain. One hypothesis is that the CR and IOR act as a scaffold to tether the ovary to the mesonephros and allow integration of the mesovarium and other extrinsic components into the ovarian medulla as the ovary folds. In addition, the cells of the CR and/or IOR may produce guidance molecules that promote the invasion of extrinsic tissues into the ovary. The EOR develops inside the ovarian capsule in fetal life, but later resides in the periovarian fat pad in postnatal mice, leading to the hypothesis that the fetal ovarian capsule may give rise in postnatal life to both the ovarian bursa and the periovarian fat pad. While the absence of the EOR, and/or CR/IOR at E17.5 did not lead to overt differences in ovary folding, more experiments with these genetic models are required to investigate exactly when during development loss of the RO occurs in these models, and whether the loss of these regions affects folliculogenesis or the internal architecture of the ovary (*Torres et al., 1995*). Importantly, while the RO is often dismissed as a vestigial remnant of the mesonephric tubules believed to degenerate in adulthood, we show that all three regions of the RO are maintained in the post-pubertal mouse ovary, and that the EOR remains a prominent structure that resides in the periovarian fat tissue. Taken together, these observations highlight the necessity of better understanding the biological function of the RO, and its role in female fertility.

While the timing of ovarian cell differentiation varies greatly across species, the regionalization of the ovarian domain into cortex and medulla is a highly conserved feature, not only across mammalian

species, but also in reptilian and avian species (*DeFalco and Capel, 2009*; *Nicol et al., 2022*). Similar to mice, the ovarian cortex of large mammals such as cows and sheep is first distinguishable by the presence of ovigerous cords and stroma, while the medulla is initially composed of stromal cells, vasculature, and IOR ovarii. The ovarian medulla then hosts the first growing follicles, while the quiescent ovarian reserve is established in the cortex (*Hummitzsch et al., 2013*; *Juengel et al., 2002*). Similar patterns have been described for many mammalian ovaries, including rabbits (*Daniel-Carlier et al., 2013*), elephants (*Stansfield et al., 2012*), and bats (*Komar et al., 2007*). It is unclear however, whether folding of the ovarian domain during fetal life could be involved in the functional regionalization of the ovary across mammalian species. To the best of our knowledge, the CSL is conserved throughout mammals (*van der Schoot and Emmen, 1996*). Images in the literature suggest that the process of ovary folding may occur at least to some degree in most studied mammalian species, including humans (*Soygur et al., 2022*). An interesting hypothesis is that the degree and direction of ovary folding could vary across species in correlation with differences in the development and final location of the infundibulum of the oviduct. Systematic comparative 3D morphometric studies of the ovary, oviduct, and CSL in various species will be required to test this hypothesis, and could lead to the identification of conserved structural changes that may be critical for the establishment of the mammalian ovarian reserve.

## Materials and methods

**Key resources table**

| Reagent type (species) or resource | Designation | Source or reference | Identifiers | Additional information |
|---|---|---|---|---|
| Strain, strain background (*Mus musculus*) | Crl:CD1(ICR) | Charles River | Strain code: 022 | |
| Strain, strain background (*M. musculus*) | C57BL/6J | Jackson Laboratory | Stock #:000664 | |
| Genetic reagent (*M. musculus*) | B6.129P2-Lgr5tm1(cre/ERT2)Cle/J | Jackson Laboratory | Stock #:008875 | |
| Genetic reagent (*M. musculus*) | Tg(Nr5a1/EGFP)1Klp | PMID:12351700 | MGI:5493455 | |
| Genetic reagent (*M. musculus*) | Tg(Wnt7a-EGFP/cre)#Bhr/Mmjax | PMID:20974921 | MMRRC Strain #036637-JAX | |
| Genetic reagent (*M. musculus*) | Pax8tm1Rdl; Pax2tm1a(KOMP)Wtsi | PMID:32381599 | | |
| Antibody | AMH (goat polyclonal) | Santa Cruz Biotechnology | #sc-6886 | 1:500 |
| Antibody | Smooth muscle alpha action (aSMA) (Cy3-conjugated mouse monoclonal) | Sigma-Aldrich | C6198 | 1:1000 |
| Antibody | Smooth muscle alpha action (aSMA) (FITC-conjugated mouse monoclonal) | Sigma-Aldrich | F3777 | 1:500 |
| Antibody | E-Cadherin (rat monoclonal) | Zymed (Thermo Fisher Scientific) | 13-1900 | 1:500 |
| Antibody | Endomucin (rat monoclonal) | Santa Cruz Biotechnology | sc-65495 | 1:500 |
| Antibody | FOXL2 (goat polyclonal) | Novus Biologicals | NB-100-1277 | 1:250 |
| Antibody | GATA4 (goat polyclonal) | Santa Cruz Biotechnology | sc-1237 | 1:250 |
| Antibody | GFP (chicken polyclonal) | Abcam | ab13970 | 1:1000 |
| Antibody | HuC/D (human polyclonal) | Gift from V. Lennon (Mayo Clinic) | | 1:10,000 |
| Antibody | KRT8 (rat monoclonal) | DSHB | TROMA-I | 1:250 |

*Continued on next page*

*Continued*

| Reagent type (species) or resource | Designation | Source or reference | Identifiers | Additional information |
|---|---|---|---|---|
| Antibody | PAX8 (rabbit polyclonal) *This antibody has cross-reactivities with other PAX family members* | Proteintech; *Moretti et al., 2012* | A10336-1-AP | 1:500 |
| Antibody | RUNX1 (rabbit monoclonal) | Abcam | ab92336 | 1:500 |
| Antibody | SOX9 (rabbit polyclonal) | Millipore | AB5535 | 1:1000 |
| Antibody | TNC (rabbit polyclonal) | Gift from H. Erickson (Duke University) | | 1:250 |
| Antibody | TUJ1 (rabbit polyclonal) | Abcam | ab18207 | 1:1000 |
| Antibody | AF647 anti-Rabbit (donkey polyclonal) | Jackson ImmunoResearch | 711-605-152 | 1:500 |
| Antibody | AF488 anti-Chicken (donkey polyclonal) | Jackson ImmunoResearch | 703-545-155 | 1:500 |
| Antibody | AF488 anti-Human (donkey polyclonal) | Jackson ImmunoResearch | 709-545-149 | 1:500 |
| Antibody | AF488 anti-Rat (donkey polyclonal) | Life Technologies | A-21208 | 1:500 |
| Antibody | Cy3 anti-Goat (donkey polyclonal) | Jackson ImmunoResearch | 705-165-147 | 1:500 |
| Antibody | Cy3 anti-Chicken (donkey polyclonal) | Jackson ImmunoResearch | 705-165-155 | 1:500 |
| Antibody | Cy3 anti-Rat (donkey polyclonal) | Jackson ImmunoResearch | 712-165-150 | 1:500 |
| Chemical compound, drug | CF-647-hydrazide probe (to label Elastin) | MilliporeSigma | SCJ4600046 | 1:500 |
| Chemical compound, drug | Dichloromethane | MilliporeSigma | 270997-1L | |
| Chemical compound, drug | Benzyl Ether | MilliporeSigma | 108014-1KG | |
| Chemical compound, drug | Heparin | MilliporeSigma | H3393-50KU | |
| Chemical compound, drug | Quadrol = N,N,N′,N′-Tetrakis(2-Hydroxypropyl) ethylenediamine | MilliporeSigma | 122262 | |
| Software, algorithm | Zen Black Edition | Carl Zeiss | | |
| Software, algorithm | Imaris v9.6 | Bitplane | | |
| Software, algorithm | Adobe Creative Cloud | Adobe | | Photoshop, Illustrator, Premier Pro |

## Mice

Unless otherwise stated, experiments were performed on wild-type CD-1 embryos, or mixed CD-1 and C57BL/6J embryos. The *Lgr5-Gfp-IRES-CreER* (*Lgr5-Gfp*; RRID: IMSR_JAX:008875) and Tg (*Nr5a1-eGFP*) (SF1-eGFP) lines were previously described (*Ng et al., 2014*; *Stallings et al., 2002*) and maintained on a C57BL/6 background. *Pax2* and *Pax8* ubiquitous deletion alleles (*Pax2del* and *Pax8del*) were obtained by crossing *Pax2f/f*; *Pax8f/f* (*Laszczyk et al., 2020*) to *Wnt7a-Cre* (RRID: MMRRC_036637-JAX) (*Winuthayanon et al., 2010*), which we found to cause germline deletion in our animals. Offspring from this cross thus carried germline CRE-independent deletions of *Pax2* and *Pax8* and sired F2 offspring with ubiquitous deletion of *Pax2*, *Pax8*, or both. The *Pax2/8* flox mice were maintained on a mixed CD-1/C57BL/6J genetic background, and the *Wnt7a-Cre* mice were on a predominantly C57BL/6J genetic background. Primers used for PCR genotyping are listed in *Supplementary file 1b*. To collect embryos at specific developmental stages, males were set up in timed matings with several females. Females were checked daily for the presence of a vaginal plug. Date of the plug was considered embryonic day 0.5. All mice were housed in accordance with National Institutes of Health guidelines, in a barrier facility maintained at a temperature of 22 ± 0.1°C, 30–70% humidity, within individually ventilated cages (Allentown; PNC75JU160SPCD3),

and a controlled 12 h on/12 h off light cycle. All experiments were conducted with the approval of the Duke University Medical Center Institutional Animal Care and Use Committee (IACUC protocol #A089-20-04 9N).

## Whole embryo collection and genotyping

Whole embryos at E14.5 were dissected in phosphate-buffered saline (PBS)$^{-/-}$, fixed in 4% paraformaldehyde (PFA)/PBS overnight at 4°C and gradually dehydrated using methanol dilutions for 45 min each at room temperature (RT): 25% MeOH/PBS; 50% MeOH/PBS; 75% MeOH/PBS; and 100% MeOH. Embryos were then stored at –20°C in 100% MeOH until required for staining. The amniotic sac of each embryo was used for sex genotyping: the tissue was incubated overnight at 55°C in tissue lysis buffer with Proteinase K. The following day, genomic DNA was precipitated with isopropanol and purified by centrifugation. The pellet was resuspended in sterile dH$_2$O and processed for PCR genotyping for *Sry* to determine the sex of the embryo. Primers used for PCR genotyping are listed in *Supplementary file 1b*.

## Reproductive complex collection

Ovary/mesonephros/MD complexes at desired stages were dissected in PBS$^{-/-}$, with as little removal of surrounding tissue as possible to maintain the integrity of the complex. Samples were fixed for 30 min at RT in 4% PFA/PBS. Following two 15 min washes, samples were gradually dehydrated in MeOH dilutions (25% MeOH/PBS; 50% MeOH/PBS; 75% MeOH/PBS; and 100% MeOH) for 15 min each at RT. Samples were stored at –20°C in 100% MeOH until required for staining.

## iDISCO+ clearing and immunostaining for lightsheet imaging of whole embryos

When ready for analysis, E14.5 XX embryos were incubated overnight in, 66% dichloromethane (DCM), 33% methanol rocking at RT. Samples were treated with 5% H$_2$O$_2$ in methanol overnight at 4°C. After washes in a reverse MeOH/PBS gradient (75% MeOH/PBS; 50% MeOH/PBS; and 25% MeOH/PBS), rehydrated tissues were washed 15 min in PBS 0.2% Triton X-102 (PTx.2), and permeabilized overnight at 37°C in iDISCO permeabilization solution (PTx.2, 2.3% glycine, 20% DMSO). The following day, samples were transferred to iDISCO blocking solution (PTx.2 10% DMSO, 1.5% horse serum) for 6 hr at 37°C. Tissues were incubated in primary antibodies diluted in PTwH (PTx.2 with 0.001% heparin) 3% horse serum, 10% DMSO for 72 hr at 37°C. Samples were washed three times for 1 hr in PTwH and incubated for 48 hr in secondary antibodies diluted in PTwH 3% horse serum at 37°C. After three 1 hr washes in PtWH, embryos were transferred to glass scintillation vials and progressively dehydrated into 100% methanol then incubated for 3 hr in 66% DCM, 33% methanol rocking at RT. Following two 30 min washes in 100% DCM, samples were transferred into 100% dibenzylether (DBE) for final clearing. All primary antibodies and fluorophore-conjugated secondary antibodies used in this study are listed in *Supplementary file 1c and d* respectively.

## Immunostaining for confocal imaging

Samples were gradually rehydrated into PBS through 10 min washes in a reverse methanol gradient (75% MeOH/PBS; 50% MeOH/PBS; and 25% MeOH/PBS), and transferred to PBS 0.1% Triton X-100 for 30 min. Samples were then transferred to blocking solution (PBS; 1% Triton X-100; 10% horse serum) for 1 hr, and incubated overnight in primary antibodies diluted in blocking solution at 4°C (*Supplementary file 1c*). The next day, samples were washed three times for 30 min in PBS 0.1% Triton X-100 and incubated overnight in secondary antibodies and Hoechst vital dye diluted 1:500 in blocking solution (*Supplementary file 1d*). On day 3, samples were washed two times for 15 min in PBS 0.1% Triton X-100 and transferred to PBS at 4°C until ready to mount for confocal imaging (1–48 hr). Samples were mounted for imaging in polyvinyl alcohol mounting solution (stored at 4°C) or DABCO mounting solution (stored at –20°C) and stored until imaged. To image the dorsal and ventral sides of the same sample, we used 3D-printed reversible slides that can be flipped and imaged from both sides. The 3D model is available for download on the NIH 3D Print Exchange website at https://3dprint.nih.gov/discover/3DPX-009765. Angles of view for confocal microscopy are determined by the angle the tissue is mounted at, and thus may differ slightly from those chosen for 3D images captured with lightsheet microscopy.

## iDISCO+CUBIC clearing and immunostaining for lightsheet imaging

Fetal reproductive complexes were immunostained and cleared with our combined iDISCO+CUBIC clearing method to allow imaging with the Zeiss Z.1 Lightsheet, which was only compatible with aqueous-based clearing solutions (*McKey et al., 2020*; *McKey et al., 2019*). Briefly, isolated ovary/ mesonephros samples went through the steps described above for iDISCO clearing. After incubation in DBE overnight, samples were transferred to 100% MeOH and progressively rehydrated to PBS through a reverse methanol gradient. Samples were then transferred into a 1:1 dilution of CUBIC Reagent-1 in $dH_2O$ (1/2Cubic R-1) for overnight incubation at 37°C. The next day, samples were transferred into 100% CUBIC R-1 for 24–48 hr. When sufficiently cleared, samples were washed in PBS and transferred to a 1:1 dilution of CUBIC Reagent-2 in PBS for 6 hr and 24–48 hr at 37°C in 100% Cubic R-2 containing Hoechst nuclear dye. Once sufficiently cleared, samples were embedded in 2% agar/ CUBIC R-2 in 1 mL syringes and stored in the dark at RT until imaged. Samples were imaged in CUBIC reagent-2 using the Z.1 Zeiss Lightsheet microscope (Carl Zeiss, Inc, Germany).

## Image acquisition

We used confocal microscopy to image whole-mount immunostained gonads used for analysis of protein expression domains, and lightsheet microscopy when our goal was to generate 3D models of the developing embryo and/or ovary/mesonephros complexes. Images of the E14.5 iDISCO+-cleared whole embryo were acquired using a LaVision BioTec Ultramicroscope II lightsheet microscope (LaVision BioTec, GmbH, Germany) equipped with zoom body optics, an Andor Neo sCMOS camera, an Olympus MVPLAPO 2×/0.5 objective and a 5.7 mm working distance corrected dipping cap (total magnifications ranging from 1.3× to 12.6× with variable zoom). The whole embryo was mounted in a sample holder and submerged in a 100% DBE reservoir. The sample was imaged at 1.26× (0.63× zoom), using the three lightsheet configuration from both sides with the horizontal focus centered in the middle of the field of view, an NA of 0.026 (beam waist at horizontal focus=28 μm), and a lightsheet width of 90%. Spacing of Z-slices was 10 μm. Confocal images of uncleared fetal ovary/mesonephros complexes were captured in the longitudinal plane on Zeiss LSM710 or LSM880 confocal microscopes and the affiliated Zen software (Carl Zeiss, Inc, Germany) using a 10× objective. Images of iDISCO+CUBIC-cleared isolated ovary/mesonephros complexes were acquired using a Zeiss Z.1 Lightsheet (Carl Zeiss, Inc, Germany). Lightsheet images were collected using the 5× NA 0.16 dry objective. Images were captured at 0.5–1× zoom, using the dual or single side lightsheet illumination (depending on size of sample), a lightsheet thickness of 2.5 μm, and a Z-interval of 1.2–2 μm. Image acquisition was performed with Zen software (Carl Zeiss, Inc, Germany).

## Image processing

Confocal Z-stacks were imported into FIJI software for minor processing, including cropping and rotations. All images of individual ovary/mesonephros complexes were oriented with the anterior end of the gonad to the left and posterior end to the right. Unless otherwise stated, all images presented in this study are maximum intensity projections of confocal Z-stacks generated with FIJI. Final processing, such as channel overlays, brightness, and contrast adjustments, and figure compositions were made in Adobe Photoshop CC (Adobe, Inc, CA). Lightsheet Z-stacks were converted into.ims files using the Imaris file converter and imported into Imaris software for processing (Version 9.6; Bitplane, Inc, UK). Upon opening the file for the first time, x and y dimensions were downsampled to 50% of the original size (1920×1920×1.2 px to 960×960×1.2 px) for better file management. 3D volumes were rendered using the *maximum intensity* render mode, and isosurfaces were generated using the *surface creation wizard*, using an absolute threshold specified for each image and channel. To display specific regions and tissues, the *clipping plane* tool was applied. In cases where the antibodies labeled more than the region desired for display, the *cut surface* tool was used. Each channel was individually adjusted for brightness and contrast to capture still images of native data or 3D models using the *snapshot* tool (images captured as fixed size 1000×1000 px transparent PNGs). Videos were captured using the *animation* tool in Imaris and edited using Adobe Premiere Pro CC software (Adobe, Inc, CA).

## Acknowledgements

The authors would like to thank Benjamin Carlson and Lisa Cameron from the Duke Light Microscopy Core Facility for confocal and lightsheet imaging resources and assistance on lightsheet imaging and analysis. The authors are grateful to all members of the Capel laboratory for helpful discussions and suggestions on the work presented here, especially to Jordan Batchvarov for outstanding mouse care and technical support. The authors thank Dr. Vanda Lennon from the Mayo Clinic (Rochester, MN) for the HuC/D antibodies, Dr. Harold Erickson (Duke University) for the TNC antibodies, and Greg Dressler for generating the *Pax2* and *Pax8* flox mice, and making them freely available to us through RB. This project was supported by a grant from the National Institutes of Health (Grant 1R01HD090050-0) (to BC) and R37HD030284 and Ben F Love Chair to RRB. CB was supported by a grant from the National Institutes of Health (Grant R37HD039963). JM was supported by postdoctoral fellowships from the Fondation ARC pour la Recherche contre le Cancer (Award #SAE20151203560) and from the American Cancer Society (Award #130426-PF-17-209-01-TBG). AEO was supported by a Diversity Supplement to HD30284. Lightsheet microscopy was supported by a grant from the National Institutes of Health (Grant #1S10OD020010-01A1) to the Duke Light Microscopy Core Facility and by a voucher from the Duke University School of Medicine.

## Additional information

### Funding

| Funder | Grant reference number | Author |
|---|---|---|
| National Institutes of Health | 1R01HD090050-0 | Dilara N Anbarci<br>Blanche Capel |
| National Institutes of Health | R37HD030284 | Alejandra E Ontiveros<br>Richard R Behringer |
| National Institutes of Health | R37HD039963 | Corey Bunce<br>Blanche Capel |
| National Institutes of Health | K99HD103778 | Jennifer McKey |
| American Cancer Society | 130426-PF-17-209-01-TBG | Jennifer McKey |
| National Institutes of Health | 1S10OD020010-01A1 | Jennifer McKey |
| Ben F Love Chair | | Richard R Behringer |
| Fondation ARC pour la Recherche sur le Cancer | SAE20151203560 | Jennifer McKey |
| National Institute on Aging | Diversity Supplement to HD30284 | Alejandra E Ontiveros |
| Duke University | School of Medicine voucher | Blanche Capel |

The funders had no role in study design, data collection and interpretation, or the decision to submit the work for publication.

### Author contributions

Jennifer McKey, Conceptualization, Data curation, Formal analysis, Validation, Investigation, Visualization, Methodology, Writing – original draft, Writing – review and editing; Dilara N Anbarci, Formal analysis, Validation, Investigation, Visualization, Writing – review and editing, Writing - contribution to original draft; Corey Bunce, Conceptualization, Investigation, Methodology, Writing – review and editing; Alejandra E Ontiveros, Resources, Methodology; Richard R Behringer, Resources, Funding acquisition, Methodology, Writing – review and editing; Blanche Capel, Conceptualization, Resources, Supervision, Funding acquisition, Project administration, Writing – review and editing

## Author ORCIDs

Jennifer McKey (iD) http://orcid.org/0000-0002-2640-1502
Blanche Capel (iD) http://orcid.org/0000-0002-6587-0969

## Ethics

This study was performed in strict accordance with the recommendations in the Guide for the Care and Use of Laboratory Animals of the National Institutes of Health. All experiments were conducted with the approval of the Duke University Medical Center Institutional Animal Care and Use Committee (IACUC protocol # A089-20-04 9N).

## Decision letter and Author response

Decision letter https://doi.org/10.7554/eLife.81088.sa1
Author response https://doi.org/10.7554/eLife.81088.sa2

---

# Additional files

## Supplementary files

• Supplementary file 1. Supplementary tables in support of this manuscript. (a) Color-coded table recapitulating ovarian and Müllerian duct phenotypes in the Pax2 and Pax8 allelic series (green with ✓, intact; yellow with ~, perturbed; red with X, absent). (b) Table recapitulating the DNA forward (middle) and reverse (right) primers used for genotyping the transgenic mouse lines used in the present study (right). (c) Table recapitulating the primary antibodies used in this study. Information in the table columns includes, from left to right protein recognized, host species, dilution, source, and product #. (d) Table recapitulating the secondary antibodies used in this study. Information in the table columns includes, from left to right protein recognized, dilution, source, and product #.

• MDAR checklist

## Data availability

All data generated or analyzed during this study are included in the manuscript and/or supplementary materials.

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
