## [Editor Report]

This work presents a very detailed study on the morphological processes governing mouse ovary morphogenesis from E14.5 to birth using recently developed methods. As a reference to ovary morphogenesis and the establishment of the female genital tract, the manuscript is of interest to all developmental biologists as it will serve as a reference to whole embryo morphogenesis, in particular vertebrate ovary morphogenetics processes.

---

## [Decision Letter]

**Decision letter after peer review:**

Thank you for submitting your article "Integration of mouse ovary morphogenesis with developmental dynamics of the oviduct, ovarian ligaments, and rete ovarii" for consideration by *eLife*. Your article has been reviewed by 2 peer reviewers, one of whom is a member of our Board of Reviewing Editors, and the evaluation has been overseen by Didier Stainier as the Senior Editor. The reviewers have opted to remain anonymous.

Essential revisions:

1) The authors should mention somewhere in the manuscript that the antibody against PAX8 from Proteintech has cross-reactivities with the other PAX family members (as described by the manufacturer) especially PAX2, even if this does not change the conclusions of the experiments using this antibody.

2) To increase the impact of the results, the authors should provide a final figure graphically representing what they show in their study. Indeed, for readers non-specialists in the female tract, the amount of results is huge and difficult to conceptualize. Therefore, we suggest to the authors add a figure with a diagram (as simple as possible) summarizing their work at the end of the manuscript.

3) We suggest the authors enlarge the discussion to other mammals.

*Reviewer #1 (Recommendations for the authors):*

The work is mainly descriptive and of importance to ovary morphogenesis and developmental dynamics. It reveals novel relationships between the ovary and surrounding tissues. It gains from detailed microscopy and reconstruction analysis and limited genetic analysis. It paves the way for further studies and functional investigations.

As it stands, it would gain from a more cellular detailed analysis of the different morphogenetic stages. Events such as folding and elongation have been shown to rely in cell shape changes in many other morphogenetic events. Current whole organ analysis does not allow the observation of individual cells. Although it is understandable that this can not be done in all panels, it would add to the current study if the authors could provide panels with more magnification showing overall cell shape where morphogenetic events require expansion (for example during Mullerian duct expansion) and during ovary folding.

*Reviewer #2 (Recommendations for the authors):*

I have only a few recommendations for the authors:

The authors may mention somewhere in the manuscript that the antibody against PAX8 from Proteintech has cross-reactivities with the other PAX family members (as described by the manufacturer) especially PAX2, even if this does not change the conclusions of the experiments using this antibody.

To increase the impact of the results, the authors should provide a final figure graphically representing what they show in their study. Indeed, for readers non-specialists in the female tract, the amount of results is huge and difficult to conceptualize. Therefore, I suggest to the authors add a figure with a diagram (as simple as possible) summarizing their work at the end of the manuscript.

Could it be possible to enlarge a little bit the discussion to other mammals?

---

## [Author Response]

Essential revisions:1) The authors should mention somewhere in the manuscript that the antibody against PAX8 from Proteintech has cross-reactivities with the other PAX family members (as described by the manufacturer) especially PAX2, even if this does not change the conclusions of the experiments using this antibody.

We thank the reviewers for pointing this out. We have now added the following statement from Proteintech’s website “Proteintech's 10336-1-AP rabbit anti-PAX8 antibody has cross-reactivities with other PAX family members'' in the Key Resources table, and in the main text, with a citation to the article that revealed the cross-reactivity with PAX5 (Moretti et al., Mod Pathol 2012).

2) To increase the impact of the results, the authors should provide a final figure graphically representing what they show in their study. Indeed, for readers non-specialists in the female tract, the amount of results is huge and difficult to conceptualize. Therefore, we suggest to the authors add a figure with a diagram (as simple as possible) summarizing their work at the end of the manuscript.

We thank the reviewers for this helpful suggestion. We have added a summary diagram in the main text (Figure 8), which recapitulates our main findings and hypotheses. This figure is referenced in the discussion (lines 422-447), in the section describing our novel perspectives on ovary morphogenesis.

3) We suggest the authors enlarge the discussion to other mammals.

This is a great suggestion. We have broadened the scope of the discussion to ovarian morphology in other mammalian species (lines 484-501).

Reviewer #1 (Recommendations for the authors):The work is mainly descriptive and of importance to ovary morphogenesis and developmental dynamics. It reveals novel relationships between the ovary and surrounding tissues. It gains from detailed microscopy and reconstruction analysis and limited genetic analysis. It paves the way for further studies and functional investigations.As it stands, it would gain from a more cellular detailed analysis of the different morphogenetic stages. Events such as folding and elongation have been shown to rely in cell shape changes in many other morphogenetic events. Current whole organ analysis does not allow the observation of individual cells. Although it is understandable that this can not be done in all panels, it would add to the current study if the authors could provide panels with more magnification showing overall cell shape where morphogenetic events require expansion (for example during Mullerian duct expansion) and during ovary folding.

We agree with the reviewer that investigating the dynamics of cell shape changes in the ovarian domain could provide important insights into the mechanisms controlling ovary morphogenesis. However, these experiments would require a significant amount of troubleshooting to find markers compatible with tissue clearing and lightsheet imaging, and to optimize the right methods for high resolution imaging and analysis of these components in 3D. So, while we agree that this is an important area of investigation, it is beyond the scope of the current manuscript and will be reserved for future studies.